# SACP: Spatially-Adaptive Conformal Prediction in Uncertainty Quantification of Medical Image Segmentation

**Jacqueline Isabel Bereska** [1,2]                               J.I.BERESKA@AMSTERDAMUMC.NL

**Hamed Karimi** [3]                                              HAMED.KARIMI@TORONTOMU.CA

**Reza Samavi** [3,4]                                             SAMAVI@TORONTOMU.CA

[1] *Department of Radiology and Nuclear Medicine, Department of Biomedical Engineering and Physics, Amsterdam UMC, University of Amsterdam,* [2] *Cancer Center Amsterdam, Amsterdam, Netherlands*
[3] *Department of Electrical, Computer, and Biomedical Engineering, Toronto Metropolitan University,* [4] *Vector Institute, Toronto, Ontario, Canada*

**Editors:** Accepted for publication at MIDL 2025

## Abstract

While Conformal Prediction provides statistical coverage guarantees, existing non-conformity measures fail to account for spatially varying importance of predictive uncertainty in medical image segmentation. In this paper, we incorporate spatial context near critical interfaces such as a vessel or critical organ in medical image segmentation. Our framework consists of three key components: (1) a base non-conformity score derived from segmentation model probabilities, (2) employing class-conditional calibration followed by a validation mechanism equipped with a distance-weighted scoring function that exponentially decays with distance from key interfaces, and (3) a prediction set construction method that preserves coverage guarantees while providing targeted uncertainty quantification in critical regions. While our approach is generalizable to different scenarios, for validation purposes, we employ tumor segmentation in pancreatic adenocarcinoma imaging from multiple medical centers. Results demonstrate that our method achieves the desired coverage levels while generating prediction sets that adaptively expand near critical interfaces.

**Keywords:** Uncertainty, Conformal Prediction, Medical Image Segmentation.

## 1. Introduction

Distribution-free uncertainty quantification in computer vision has seen significant advances through Conformal Prediction (CP), which transforms an algorithm's predictions into prediction sets with robust finite-sample validity (Fontana et al., 2023). While CP has shown promising results in classification and regression tasks (Vazquez and Facelli, 2022; Lu et al., 2022), its application to image segmentation presents several challenges, particularly in scenarios where prediction accuracy requirements vary across the output space (Zhou et al., 2024). Standard CP approaches employ uniform non-conformity scores across prediction regions, implicitly assuming homogeneous uncertainty throughout the prediction space. However, this assumption breaks down in segmentation tasks where different spatial regions demand distinct levels of certainty measurements. For instance, boundary regions of a segment often require substantially different uncertainty characterization than interior regions. This limitation becomes particularly crucial when medical imaging is used for surgery planning and the boundary regions of a canonical object (e.g., a tumor) and its

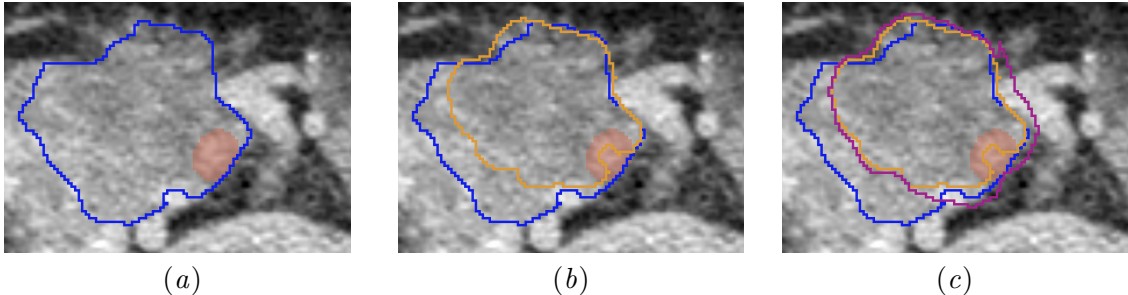

$$(a) \qquad\qquad (b) \qquad\qquad (c)$$

Figure 1: Illustration of spatially-adaptive conformal prediction for PDAC tumor segmentation: **(a)** the base segmentation of the tumor (blue) adjacent to a major vessel (red), where precise delineation of the tumor-vessel interface is crucial for surgical planning; **(b)** the base segmentation of the tumor (blue) and the standard conformal prediction set (orange) **(c)** a comparison between standard CP sets (orange) and SACP sets (purple), where our approach adapts the prediction bounds based on proximity to critical structures.

closeness to some critical masses (e.g., a vessel) requires varying confidence requirements as described in the following clinical setting.

In pancreatic ductal adenocarcinoma (PDAC) diagnosis and treatment planning, accurate tumor segmentation near critical vascular structures can mean the difference between an operable and inoperable assessment. When a tumor interfaces with major blood vessels, surgeons require millimeter-scale accuracy (with high certainty) in boundary delineation to determine resectability and surgical plans. Conversely, the precise boundary definition of the tumor's interior regions, while important, permits more flexibility in uncertainty quantification. As shown in Figure 1(a), the initial segmentation shows a tumor-vessel interface in a CT scan, where a PDAC tumor boundary (blue) is adjacent to a major vessel (red). To consider uncertainty in the prediction, one may apply CP (e.g., with error rate $\alpha = 10\%$) to ensure at least 90% certainty around the delineated boundaries of the tumor (yellow) as shown in Figure 1(b). CP generates homogeneous uncertainty set across all voxels, while for the surgical planner it's crucial to know, with higher precision (e.g., with error rate only $\alpha = 5\%$) around the vessels, due to its critical impact on surgery, and error rate $\alpha = 10\%$ elsewhere. This is the challenge the classical CP lacks addressing.

In this paper, we propose *Spatially-Adaptive Conformal Prediction* (SACP), to address the challenge of incorporating varying spatial importance of voxels into a conformal prediction set while preserving its theoretical guarantees. We introduce locally adaptive non-conformity scores that explicitly account for distance to critical structures, enabling CP to be more conservative near critical boundaries while remaining CP-level uncertain elsewhere. When SACP is applied to our motivating example, as shown in Figure 1(c), the prediction set (shown in purple) expands only around the vessel (the critical interface). This varying conservativeness directly impacts surgical decision-making and may lead the surgeon to seek further and more accurate diagnostic imaging or more precise segmentation for that specific region. Our framework consists of three key components: (1) a base non-conformity score derived from segmentation model probabilities, (2) employing class-conditional calibration followed by a validation mechanism equipped with a distance-weighted scoring function based on spatial proximity to critical interfaces, and (3) a prediction set construction method that maintains original CP coverage guarantees.

**Related Work.** Conformal Prediction (CP) provides a distribution-free framework for uncertainty quantification in machine learning, ensuring finite-sample coverage guarantees regardless of data distribution (Fontana et al., 2023; Karimi and Samavi, 2023; Zhou et al., 2024). In medical imaging, CP has been used for uncertainty estimation in diagnostic tasks (Angelopoulos et al., 2020; Karimi and Samavi, 2024) and extended to segmentation (Brunekreef et al., 2024), post-hoc uncertainty quantification (Mossina et al., 2024), and performance range prediction (Wundram et al., 2024). (Sadinle et al., 2019) introduced least ambiguous set-valued classifiers that optimize prediction set size while maintaining class-conditional coverage, but their approach is limited to multi-class classification without spatial calibration. Class-conditional CP, as formulated by (Shi et al., 2013), balances per-class coverage but lacks spatial awareness, crucial for anatomical segmentation. (Mao et al., 2024) proposed a model-free spatial prediction framework for geospatial data, yet its environmental focus limits medical imaging applicability. Similarly, (Guan, 2023) introduced localized CP that adapts to local features but does not incorporate anatomical structures. Extensions of CP beyond exchangeability address distributional shifts (Barber et al., 2023) but neglect spatial heterogeneity and pixel imbalance. Region-based CP methods (Fischer et al., 2024) offer robust uncertainty quantification for radiotherapy but lack fine-grained voxel-level adaptation. Spatial CP has been explored in remote sensing (Liu et al., 2024), yet challenges persist in medical segmentation, particularly balancing spatial context with class imbalance near critical structures. Minority class under-coverage remains a key issue, as small structures are often sparsely represented. Our approach integrates class-conditional calibration with spatial weighting using distance-decaying functions to maintain coverage while improving precision in critical regions.

**Contributions.** First, we develop a novel spatially-adaptive CP method, which is sensitive to the distance to critical structures, allowing a rigorous presentation of nonuniform uncertainty across voxels of a segment in an image. Our approach is generalizable to other fields of image segmentation where nonuniform uncertainty quantification is needed –e.g., safety in robotics and quality control in manufacturing, where a false negative can lead to accidents. Second, we theoretically prove that SACP maintains the original CP coverage, set size and adaptability characteristics. Third, we experimentally demonstrate that our approach is efficient and clinically insightful for a real-world clinical case. The code for experimental evaluation is publicly available at https://github.com/tailabTMU/SACP.

## 2. Spatially-Adaptive Conformal Prediction

**Conformal Prediction.** Conformal prediction is a statistical framework that produces prediction intervals for any underlying pretrained model with a guarantee on the prediction's reliability (Vovk et al., 2005). For a given significance level $\alpha \in \mathbb{R}^{(0,1)}$, CP ensures that for a calibration dataset $\mathcal{D} = \left\{(x_i, y_i)\right\}_{i=1}^{n}$ and a new test point $(x_{n+1}, y_{n+1})$ drawn from the same distribution, $\mathbb{P}\left(y_{n+1} \in \mathcal{C}(x_{n+1})\right) \geq 1 - \alpha$. CP defines a non-conformity score $S : \mathcal{X} \times \mathcal{Y} \to \mathbb{R}^+$ that quantifies how well $x_{n+1}$ *conforms* to the calibration dataset. The prediction set is then computed based on the empirical quantiles of these non-conformity scores: For a chosen confidence level $1-\alpha$, the prediction set $\mathcal{C}(x_{n+1})$ is defined as $\mathcal{C}(x_{n+1}) = \left\{\hat{y} \in \mathcal{Y} : S(x_{n+1}, \hat{y}) \leq \tau_\alpha\right\}$, where $\tau_\alpha$ is the $(1 - \alpha)$-quantile of the non-conformity scores. This guarantee is unconditional and holds for any model and any distribution as long as the

underlying exchangeability assumption is satisfied. The exchangeability assumption implies that for any permutation $\pi$ of $\{1, 2, \ldots, n\}$, permutations of the dataset have the same joint distribution as $\mathbb{P}\big((x_1, y_1), \ldots, (x_n, y_n)\big) = \mathbb{P}\big((x_{\pi(1)}, y_{\pi(1)}), \ldots, (x_{\pi(n)}, y_{\pi(n)})\big)$. We refer to (Angelopoulos et al., 2023) for a more in-depth introduction to CP. The *conservativeness* of this guarantee is also adjustable (by defining different thresholds for $p$-values) and can be beneficial when cautious coverage is preferred (Appendix D).

## 2.1. Problem Setup

Let $\mathcal{X} \subset \mathbb{R}^3$ represent a discretized volumetric image obtained from axial slices, where $\mathcal{X}$ is subdivided into a finite, structured grid of cuboidal units called voxels. We define each voxel $x \in \mathcal{X}$ by its indices $(x_a, x_c, x_s)$ along the axial, coronal, and sagittal axes, respectively. Considering a set of possible labels $\mathcal{Y}$, we represent the true label $y \in \mathcal{Y}$ as the indicator of the organ that the voxel $x$ belongs to, and the baseline segmentation model $f_\Theta$ obtains predictive probabilities $p(\hat{y}|x)$ associated with each label $\hat{y} \in \mathcal{Y}$.

**Definition 1 (Canonical Object)** *Label $l \in \mathcal{Y}$ denotes a canonical object, if $l$ represents a primary structure of interest for the downstream task.*

**Definition 2 (Critical Masses)** *$\mathcal{M}$ is a set of critical masses in the volume, if the proximity of any $m \in \mathcal{M}$ to the canonical object necessitates conservative decision-making for the downstream task.*

For the clinical settings described in Section 1, a tumor is a *canonical object* for the downstream task of surgery planning for the removal of that tumor and the set of vessels in the volume are *critical masses*, as when a tumor has a vessel in its proximity, the surgeon needs a prediction set with higher uncertainty (more conservative prediction).

## 2.2. Spatially-Adaptive Non-Conformity Score

To apply CP on voxel-wise tasks (e.g., tumor segmentation for surgery planning), we need to address two challenges: (1) CP uses a single threshold $\alpha$ across all classes; thus the prediction set for rare classes will be either over or under-conservative depending on the class distribution; this is particularly crucial in the tumor segmentation task as tumors are small structures relative to the total CT image volume. (2) The prediction set is invariant to the voxels, while we expect, the prediction set to be more conservative when voxels of the canonical object (e.g., tumor) is closer to one or more critical masses (e.g., vessels).

To address the first challenge, we adopt Class-Conditional Conformal Prediction (CCCP), where CP is refined to use various quantile thresholds across different classes (Shi et al., 2013; Sadinle et al., 2019). We compute a distinct threshold $\tau_\alpha^{\hat{y}}$ for each label $\hat{y} \in \mathcal{Y}$ independently as the $(1 - \alpha)$-quantile of the non-conformity scores as,

$$\tau_\alpha^{\hat{y}} = \text{Quantile}_{1-\alpha}\Big(\{S_{\text{base}}(x_i, y_i) : y_i = \hat{y}\}_{i=1}^n\Big) \ . \tag{1}$$

For a new test data point $(x_{n+1}, y_{n+1})$, the prediction set $\mathcal{C}(x_{n+1})$ is constructed as,

$$\mathcal{C}(x_{n+1}) = \big\{\hat{y} \in \mathcal{Y} : S_{\text{base}}(x_{n+1}, \hat{y}) \leq \tau_\alpha^{\hat{y}}\big\} \ . \tag{2}$$

To address the second challenge, we define a new score function, $S_{\text{SACP}}$, that augments the original CP non-conformity score function, $S_{\text{base}}$, with a parameterized weight $w_v \in \mathbb{R}^{[0.5,1)}$, as a multiplicative factor denoted as,

$$S_{\text{SACP}}(x|\hat{y} = l) = w_v(x, l) \cdot S_{\text{base}}(x|\hat{y} = l) \ , \tag{3}$$

where $v \in \mathcal{M}$ is the nearest critical mass to the voxel $x$, and $l \in \mathcal{Y}$ is the canonical object. Our intention is to make the impact of the weight irrelevant ($w_v \approx 1$) when voxels are far from both critical masses and canonical object, therefore generating a prediction set as conservative as the original base function and maximizing the impact of the weight ($w_v = 0.5$) when voxels are very close to critical masses and the canonical object. The weight has to be also impacted by our confidence in segmenting the canonical object (tumor) as well as the relevancy of the different critical masses, as we may have more than one critical mass, each with a different relevancy factor for the downstream task. Formally, we have four parameters for our weight function:

1. $\delta_m$: The Euclidean distance of each voxel $x$ to the critical mass $m \in \mathcal{M}$.

2. $\phi_l$: The Euclidean distance of each voxel $x$ to the canonical object $l \in \mathcal{Y}$.

3. $\mathcal{I}(l)$: The information-theoretical surprisal or unexpectedness of observing the canonical object $l$ with probability $p(\hat{y} = l|x)$ that inversely accounts for our confidence on correct segmentation of the canonical object, where $\mathcal{I}(l) \stackrel{\text{def}}{=} -\log p(\hat{y} = l|x)$.

4. $\gamma_m \in \mathbb{R}^{(0,1]}$: A hyperparameter capturing the relevancy and criticality of each critical mass $m \in \mathcal{M}$.

Putting all together, the weight function for each voxel $x$ is defined as,

$$w_v(x,l) = \sigma\left( \overbrace{\frac{1}{\gamma_v}\Big(\phi_l + \delta_v \mathcal{I}(l)\Big)}^{\tilde{w}_v} \right) \qquad s.t. \qquad v = \underset{m \in \mathcal{M}}{\arg\min}\ \delta_m \ , \qquad (4)$$

where $\tilde{w}_v : \mathcal{X} \times \mathcal{Y} \times \mathcal{M} \to \mathbb{R}^+$ is a function that represents the raw weight value and the constraint ensures we consider the nearest critical mass. $\tilde{w}_v$ is then normalized to $w_v(x,l) \in \mathbb{R}^{[0.5,1)}$ using the sigmoid function $\sigma(.)$ (further details in Appendix B).

The weight approaches 0.5 near critical masses or high-confidence tumor regions, increasing prediction set conservativeness. When $\delta_v$ is small and $p(\hat{y} = l|x)$ is high (i.e. $\mathcal{I}(l)$ is low), $w_v$ yields towards its lower bound, reducing the non-conformity scores of label $l$, making it more likely to be included in the prediction set (more conservative). This aligns with the desire to treat voxels around the critical masses and the canonical object more conservatively and expand the prediction set for those areas. Conversely, when $\delta_v$ is getting larger and $p(\hat{y} = l|x)$ smaller ($\mathcal{I}(l)$ is higher), $w_v$ moves closer to its upper bound of 1, eliminating the impact of the weight and making it less likely for distant regions to be included in the prediction set. $\phi_l$ also behaves similarly. Lower $\phi_l$ (the voxel being closer to the canonical object) yields lower weight, making the set more conservative and vice versa. The relevancy hyperparameter $\gamma_v$ accepts values between zero (strictly greater than zero), for the least critical mass to 1, for the most critical mass. $\gamma_v$ has a diminishing impact on $\tilde{w}_v$ and in turn to $w_v$. For example, if the user sets the relevancy for a critical mass low (e.g., $\gamma_v = 0.5$), the weight computed based on all other factors will be doubled, $w_v$ gets closer to 1 and diminishes its impact on original non-conformity score. In contrast, when relevancy increases, $w_v$ gets closer to its lower bound of 0.5 and increases the prediction set size. Note the value of $\gamma_v$ needs to be fine-tuned depending on the context of the application.

**Theorem 3 (SACP Conservativeness)** *If $S_{base}(x, \hat{y})$ denote the base non-conformity score for a voxel $x \in \mathcal{X}$ with the predictive label $\hat{y}$, and $\tau_\alpha^{\hat{y}}$ the $(1 - \alpha)$-quantile of $S_{base}$*

with the error rate $\alpha$, then for the canonical object $l \in \mathcal{Y}$, the prediction set produced by the non-conformity function $S_{SACP}(x|\hat{y} = l)$ is at least as conservative as sets produced by $S_{base}(x|\hat{y} = l)$. See Appendix C for the proof.

**Corollary 4** *For an unseen data point $x_{new}$ and $S_{SACP}(x_{new}, \hat{y})$ (Equation 3), the inclusion of canonical object label $l$ in the prediction set $\mathcal{C}(x_{new})$ with error rate $\alpha$ depends on spatial properties near high-risk regions that satisfies:*

$$l \in \mathcal{C}(x_{new}) \iff S_{SACP}(x_{new}|\hat{y} = l) \leq \tau_\alpha^l , \tag{5}$$

*where $\tau_\alpha^l$ is class-specific threshold as $(1-\alpha)$-quantile of non-conformity scores $S_{base}$.*

SACP maintains two key theoretical guarantees: (1) the standard CP coverage $\mathbb{P}(y \in \mathcal{C}(x)) \geq 1-\alpha$, and (2) for the canonical object $l$, $\mathcal{C}(x)$ is at least as conservative as standard CP set (proof in Appendix C). The spatial relationship that Equations (3) and (4) promote is particularly valuable in the clinical setting described in Section 1 for determining resectability. This assessment follows region-specific clinical guidelines - for instance, the NCCN guidelines in the US and DPCG guidelines in the Netherlands - each defining different criteria for vessel involvement and tumor contact thresholds that determine resectability. Consequently, when deploying pretrained tumor segmentation models across different regions, the calibration of vessel-specific importance factors also needs to be adjusted to align with local clinical guidelines and their specific vessel prioritization. The detailed process of generating a prediction set using SACP is described as Algorithm 1 in Appendix A.

## 3. Experimental Evaluations

**Datasets and Ground Truth.** This retrospective study analyzes 30 contrast-enhanced CT scans from the PANORAMA Challenge (Alves et al., 2024), comprising portal venous phase scans from five European centers. We further validate our approach on 30 CT scans from the Memorial Sloan Kettering (MSK) Medical Segmentation Decathlon dataset (Simpson et al., 2019), with detailed results in Appendix E. Ground truth segmentations for PDAC tumors were provided by expert radiologists, while anatomical structures (pancreas, duodenum, liver, gallbladder, kidneys, adrenal glands, spleen) were segmented using TotalSegmentator (Wasserthal et al., 2023). A hierarchical fusion strategy prioritizes radiologist annotations over automated segmentations.

**AI-Generated Segmentations.** We employ two deep learning models: a primary model for PDAC tumors and surrounding anatomical structures (Bereska et al., 2024), and a vessel-specific model focused on structures critical for PDAC resectability assessment. The latter segments five key vessels: the celiac trunk (CeTr), hepatic artery (HA), portal vein (PV), superior mesenteric artery (SMA), and superior mesenteric vein (SMV). The PDAC segmentation employs a tripartite architecture of teacher, professor, and student models, all implemented using 3D UNet cascades. The final student model was trained on a dataset of 1085 CTs from 903 patients (see Appendix F for details). For subsequent non-conformity score computation, we preserve (1) the pre-softmax probability maps for all 11 classes (10 anatomical labels plus background) from the primary segmentation model, and (2) distance maps computed from the vessel-specific model, measuring the distance from each voxel to each of the five resectability-determining vessels. To ensure robustness to outliers, both

distance and probability values are clipped at their respective 95th percentiles before being used in the non-conformity score computation. Our analysis shows these segmentation models exhibit spatially varying accuracy, with significantly lower Dice scores near vessels ($\leq$ 5mm: median 0.75, mean 0.64) compared to more distant regions (median 0.812, mean 0.75). This performance gap, coupled with higher variability near vessels (SD: 0.27 vs. 0.15), highlights a key challenge in medical image segmentation-critical regions often suffer from both lower model performance and greater inter-observer variability. Since ground truth itself is ambiguous in these areas, improving segmentation models alone may not suffice, underscoring the need for a more precise, region-sensitive uncertainty quantification.

**Cropping.** To optimize computational efficiency, we use an adaptive 3D bounding box cropping strategy. We identify the minimal volumetric boundary encompassing all voxels with specified target labels (gallbladder, pancreas, duodenum, and tumor) and apply this crop consistently across all corresponding image modalities and their derivatives.

### 3.1. Evaluation Metrics

We evaluate our spatially-adaptive framework through metrics assessing both predictive performance and anatomical sensitivity. For each voxel $x \in \mathcal{X}$ with confidence level $1 - \alpha$, we compute the empirical coverage rate for the set-valued predictor function $\mathcal{C}$ as $\text{cov}(\mathcal{C}) = \frac{1}{|\mathcal{X}|} \sum_{x \in \mathcal{X}} \mathbb{1}_{\{y \in \mathcal{C}(x)\}}$, where $\mathbb{1}$ is the indicator function and $y \in \mathcal{Y}$ is the true label. Coverage is assessed separately for vessel-adjacent ($\delta_v \leq$ 5mm) and non-critical regions ($\delta_v >$ 5mm). A higher coverage rate indicates increased conservativeness in the model's predictions, prompting clinicians to exercise additional caution and potentially seek further diagnostic imaging or expert review for regions where the prediction sets are larger.

To evaluate calibration quality across different prediction set sizes, we employ Size-Stratified Coverage Violation (SSCV) analysis, which examines how empirical coverage varies in $K$ different ranges (bins) of prediction set cardinality as $\mathcal{S}_j \subset \{1, 2, \ldots, |\mathcal{Y}|\}$ where $\mathcal{Y}$ is the set of possible classes. $\text{SSCV} \in \mathbb{R}^{[0,1)}$ is computed as $\text{SSCV}(\mathcal{C}, \{\mathcal{S}_j\}_{j=1}^{K}) = \sup_j |\frac{1}{|\mathcal{X}_j|} \sum_{x \in \mathcal{X}_j} \mathbb{1}_{\{y \in \mathcal{C}(x)\}} - (1 - \alpha)|$, where $\mathcal{X}_j = \{x \in \mathcal{X} : |\mathcal{C}(x)| \in \mathcal{S}_j\}$ represents the set of voxels having prediction sets in size range $\mathcal{S}_j$ (Angelopoulos et al., 2020). Lower SSCV values indicate better calibration across different set sizes. We also define the Relative Width Ratio (RWR) to quantify adaptation of prediction set sizes based on anatomical criticality around an arbitrary distance threshold $r$ as,

$$\rho(r) = \frac{\mu(\mathcal{C}|\delta_v > r)}{\mu(\mathcal{C}|\delta_v \leq r)} \ , \tag{6}$$

where $\mu(\mathcal{C}|\delta_v) = \frac{1}{|\mathcal{X}_v|} \sum_{x \in \mathcal{X}_v} |\mathcal{C}(x)|$ represents the average set size to evaluate prediction set efficiency for the set of voxels $\mathcal{X}_v$ at distance $\delta_v$ from the nearest vessel $v$.

### 3.2. Experimental Setup

We use 10 cases for calibration to determine class-specific non-conformity score thresholds $\tau_\alpha^{\hat{y}}$ for each label $\hat{y} \in \mathcal{Y}$ and evaluate on 20 held-out cases. Statistical comparisons use paired t-tests with Benjamini-Hochberg correction ($p < 0.05$). For the vessel-specific analysis, we incorporate anatomical context through a weighted scoring mechanism. Critical vessels are assigned different relevancy hyperparameters ($\gamma$) based on the NCCN resectability criteria for PDAC, with arterial vessels (CeTr, HA, SMA) receiving higher weights ($\gamma = 0.8$)

Table 1: Coverage and RWR analysis across vessel proximity zones using Least Ambiguous Confidence score (LAC) at $\alpha = 0.05$.

| Distance | CCCP | | SACP | |
|---|---|---|---|---|
| | Coverage | RWR | Coverage | RWR |
| $\leq$ 2mm | 0.954 | 2.887 | 0.981 | 2.762 |
| $\leq$ 5mm | 0.970 | 2.702 | 0.987 | 2.684 |
| $\leq$ 10mm | 0.977 | 2.611 | 0.988 | 2.621 |
| $\leq$ 20mm | 0.978 | 2.574 | 0.987 | 2.592 |
| > 20mm | 0.982 | 2.509 | 0.988 | 2.525 |

Table 2: Vessel-specific coverage rates at different proximity zones for CCCP (C) and SACP (S) at $\alpha = 0.05$.

| Vessel | 2mm | | 5mm | | 10mm | | 20mm | | >20mm | |
|---|---|---|---|---|---|---|---|---|---|---|
| | C | S | C | S | C | S | C | S | C | S |
| CeTr | 0.999 | 1.000 | 0.999 | 1.000 | 0.998 | 1.000 | 0.980 | 0.987 | 0.980 | 0.988 |
| HA | 0.959 | 0.980 | 0.973 | 0.986 | 0.987 | 0.994 | 0.981 | 0.989 | 0.980 | 0.987 |
| SMA | 0.925 | 0.975 | 0.967 | 0.989 | 0.982 | 0.994 | 0.973 | 0.985 | 0.984 | 0.989 |
| PV | 0.927 | 0.953 | 0.955 | 0.974 | 0.957 | 0.974 | 0.978 | 0.989 | 0.981 | 0.987 |
| SMV | 0.960 | 0.997 | 0.956 | 0.987 | 0.958 | 0.980 | 0.975 | 0.987 | 0.983 | 0.987 |

compared to venous vessels (PV, SMV: $\gamma = 0.6$). This weighting scheme reflects their relative importance in determining resectability, as arterial involvement beyond $180°$ renders a tumor unresectable, while venous involvement may permit resection with reconstruction.

To achieve sharp transitions in uncertainty estimates near vessel boundaries, we amplify the sigmoid response using a gain factor ($\beta = 10$), creating more pronounced changes in uncertainty estimates as predictions approach critical vascular structures. This enhanced sigmoid sensitivity provides a clearer delineation of high-risk regions for surgical planning. Computational requirements and performance metrics are detailed in Appendix I.

### 3.3. Experimental Results

**Coverage Analysis.** Our framework achieves strong coverage on the PANORAMA dataset ($n = 20$) with an overall coverage of 0.987 (mean per-case: $0.981\pm0.005$ SEM). The coverage significantly exceeds the target coverage of 0.95 (Wilcoxon signed-rank test, $p = 0.0007$). Size-stratified coverage violation (SSCV) analysis for the tumor label revealed consistent calibration across 89% of voxels having prediction sets of size 0-3 elements with coverage violation of 0.037 and coverage rates between 0.987 and 0.988.

**Distance-Based Analysis.** As shown in Table 1, prediction set size decreases with distance from vessels while maintaining high coverage. RWR ranges from $2.762 \pm 0.150$ SEM near vessels ($\leq$2mm) to $2.525 \pm 0.036$ SEM beyond 20mm, with coverage remaining consistently high across all distances (0.981-0.988). This decreasing RWR pattern suggests our method adapts to provide more precise predictions in regions farther from vessels, while maintaining wider prediction sets near critical vascular structures. Vessel-specific analysis in Table 2 demonstrates robust performance across all major vessels, with excellent coverage in vessel-proximate regions. Notably, we achieve high coverage in critical surgical planning zones, particularly near arteries. Visual examples of the prediction sets and their relationship to vessel proximity are provided in Appendix G. We conducted additional experiments with varying vessel relevancy hyperparameters ($\gamma$) to understand their impact on prediction set characteristics and coverage guarantees; detailed results are presented in Appendix J.

**Comparison with Standard Class-Conditional CP.** Our spatially adaptive approach demonstrates significantly improved coverage ($0.981 \pm 0.005$ SEM vs $0.968 \pm 0.038$ SEM, paired t-test t=3.366, p=0.003). Near vessels ($\leq 2mm$), we achieve both superior coverage (0.981 vs. 0.954) and reduced RWR (2.762 vs. 2.887). Figure 2 shows consistently better coverage across target confidence levels, particularly in the $40 - 80\%$ range. Our method maintains high coverage while exhibiting decreasing RWR with distance from vessels, from $2.762 \pm 0.150$ SEM at $\leq 2mm$ to $2.525 \pm 0.036$ SEM beyond 20mm, demonstrating that our framework effectively adapts prediction sets based on proximity to critical anatomical

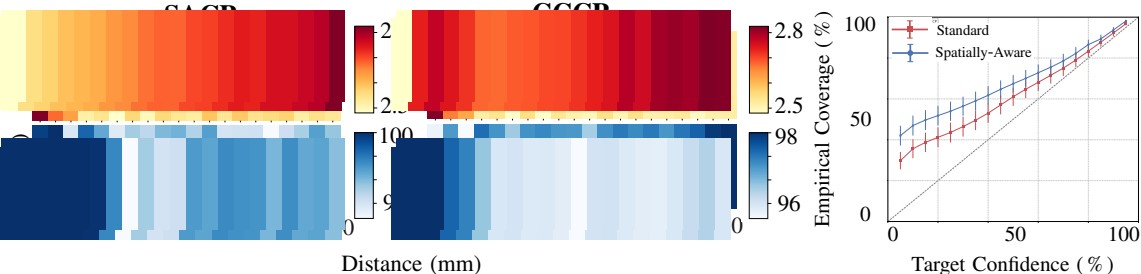

Figure 2: Left: RWR (top) and coverage (bottom) as a function of vessel distance for both datasets. Right: Comparison of empirical coverage at different confidence levels between our method (SACP) and standard Class-Conditional CP (CCCP).

structures. Results from additional experiments, including including uniformly conservative CCCP and a binary weighting scheme (Appendices K and L), and validation on the MSK dataset demonstrating generalizability across different clinical contexts (Appendix E).

## 4. Conclusion

We presented a spatially-adaptive conformal prediction framework that provides anatomically informed uncertainty quantification for medical image segmentation. Our method adapts prediction sets based on proximity to critical vascular structures while maintaining theoretical coverage guarantees. Validation on the PANORAMA dataset demonstrates robust performance, with strong coverage in vessel-adjacent regions and efficient adaptation of prediction set sizes based on anatomical criticality.

**SACP Generalizability Beyond Medical Imaging.** While SACP was motivated by and validated on pancreatic cancer surgical planning, the same principles described in this paper apply to more general scenarios where the segmentation task has to deal with (1) a spatially structured prediction task, (2) the existence of canonical objects, and (3) the existence of critical structures within the spatial domain. For example, in natural disaster prediction from satellite imagery, uncertainty in flood boundary delineation near populated areas or critical infrastructure carries greater consequences than in remote regions.

**SACP Limitations.** The method depends on critical mass segmentation and distance maps as inputs, and inter-observer variability in ground truth annotations—especially for PDAC—adds uncertainty at anatomical interfaces. The spatial weighting mechanism, though effective, requires careful hyperparameter tuning. While our analysis leverages millions of voxels, the number of distinct tumor-vessel interfaces is constrained by our sample size, limiting the diversity of spatial relationships evaluated. Despite these challenges, this approach advances AI-driven precision in surgical planning and patient care.

**Future Work.** We plan to validate our approach on larger datasets to further demonstrate the generalizability of SACP. We consider applying this framework to robotics, industrial defect detection, and disaster prediction, where boundary uncertainty affects decisions. Relaxing the exchangeability assumption to handle distribution shifts with model adaptations is another key area. Finally, exploring robust statistical techniques to maintain coverage under worst-case perturbations could enhance reliability in safety-critical applications.

## Acknowledgments

R.S. is supported by the Natural Sciences and Engineering Research Council of Canada (NSERC) Discovery grant RGPIN-2016-06062, the HHS Deep Learning grant, and the TMU start-up grant. J.B. was supported by the Cancer Center Amsterdam's Young Talents Travel Grant for a research visit to Toronto Metropolitan University. We thank Leonard Bereska for his inspiration, proofreading, and contributions to visualization.

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

## Appendix A. Algorithmic Description of SACP

In Algorithm 1, we describe the step-wise procedure and the required computation regarding applying SACP to incorporate spatial context in 3D voxel-wise segmentation and enhance uncertainty quantification.

- In Step 1, a pretrained segmentation model $f_\Theta$ generates voxel-wise predictive probabilities in an input volume $\mathcal{X}$ using the softmax function.

- In Step 2, we apply class-conditional calibration to ensure a desired confidence rate of at least $1 - \alpha$ for each class $\hat{y} \in \mathcal{Y}$ using $S_{\text{base}}$ non-conformity scores of calibration set of voxels. For each class, the $(1 - \alpha)$-quantile threshold $\tau_\alpha^{\hat{y}}$ is determined, setting the baseline for prediction set construction.

- In Step 3, we compute spatial properties as the Euclidean distances of each voxel $x \in \mathcal{X}$ to a set of critical masses $m \in \mathcal{M}$ denoted by $\delta_m$ and to the canonical object label $l \in \mathcal{Y}$ denoted by $\phi_l$.

- In Step 4, we identify the nearest critical mass $v \in \mathcal{M}$ for each voxel, forming a spatial reference. Then, a normalized weight $w_v$ is computed for each voxel $x$ based on its proximity to the canonical object $l$ and the nearest critical mass $v$, adjusted by a mass-specific relevance factor $\gamma_v$. This weight modulates the base non-conformity score $S_{\text{base}}$ associated with the canonical object $l$ to refine uncertainty estimation relative to spatial critical structures. Finally, the prediction set $\mathcal{C}(x)$ is constructed by including the canonical object label $l$ in the set if and only if the adjusted score $S_{\text{SACP}}$ remains below its respective (class-conditional) quantile threshold $\tau_\alpha^l$.

By integrating spatial information, SACP improves the reliability of conformal prediction in 3D segmentation, particularly in anatomically structured regions where spatial coherence is essential.

## Appendix B. Further Details of SACP Parameters

We compute $\delta_m$ as the Euclidean distance from voxel $x$ to any of the potential critical masses $m \in \mathcal{M}$ (e.g., major vessels) that is defined by the function $d : \mathcal{X} \times \mathcal{M} \to \mathbb{R}^+$ as,

$$\delta_m = d_{Euc}(x, m) = \min_{x' \in V_m} ||x - x'|| \, , \tag{7}$$

where $m \in \mathcal{M}$ is a critical mass containing a set of voxels $V_m \subset \mathcal{X}$.

We compute $\phi_l$ as the Euclidean distance from voxel $x$ to the segmentation outcome of a pretrained model $f_\Theta$ that is defined by the function $\hat{d} : \mathcal{X} \times \mathcal{Y} \to \mathbb{R}^+$ as,

$$\phi_l = \hat{d}_{Euc}(x, l) = \min_{x' \in V_l} ||x - x'|| \, , \tag{8}$$

where $l \in \mathcal{Y}$ is the canonical object label and $V_l \subset \mathcal{X}$ contains a set of voxels that are segmented as label $l$ such that:

$$V_l = \left\{ x' \in \mathcal{X} \mid \arg\max_{\hat{y} \in \mathcal{Y}} f_\Theta(x', \hat{y}) = l \right\} \, , \tag{9}$$

**Algorithm 1:** Spatially-Adaptive Conformal Prediction (SACP)

---

**Input:** 3D input volume $\mathcal{X}$: voxels $x$ with true labels $y$; Set of all possible labels $\hat{y} \in \mathcal{Y}$; set of critical masses $\mathcal{M}$; canonical object label $l$; pretrained segmentation model $f_\Theta$; desired error rate $\alpha$; mass-specific relevance factors $\{\gamma_m\}_{m \in \mathcal{M}}$;

**Output:** $\mathcal{C}(x)$ as prediction set for each voxel;

// Step 1: Get model predictions

1   $\forall x \in \mathcal{X}, \hat{y} \in \mathcal{Y}: \; p(\hat{y}|x) \leftarrow \text{softmax}\big(f_\Theta(x, \hat{y})\big)$      // Get predictive probabilities

// Step 2: Class-conditional calibration on $n$ voxels

2   **for** *each class* $\hat{y} \in \mathcal{Y}$ **do**

3     $\tau_\alpha^{\hat{y}} \leftarrow \text{Quantile}_{1-\alpha}\Big(\big\{S_{\text{base}}(x_i, y_i): \; y_i = \hat{y}\big\}_{i=1}^n\Big)$

4   **end**

// Step 3: Compute spatial distances

5   **for** *each voxel* $x \in \mathcal{X}$ **do**

6     $\forall m \in \mathcal{M}: \; \delta_m \leftarrow d_{Euc}(x, m)$      // Distance to the critical masses (Eq. 7)

7     $V_l \leftarrow \{x' \in \mathcal{X} \,|\, \arg\max_{\{\hat{y} \in \mathcal{Y}\}} f_\Theta(x', \hat{y}) = l\}$    // Set of canonical object voxels

8     $\phi_l \leftarrow \hat{d}_{Euc}(x, l)$         // Distance to the canonical object (Eq. 8 and 9)

9   **end**

// Step 4: Generate SACP prediction sets

10   **for** *each voxel* $x \in \mathcal{X}$ **do**

11     $v = \arg\min_{\{m \in \mathcal{M}\}} \delta_m$             // Find the nearest critical mass

12     $w_v(x, l) \leftarrow \sigma\Big(\frac{1}{\gamma_v}\big(\phi_l + \delta_v \mathcal{I}(l)\big)\Big)$       // Compute spatial weight (Eq. 4)

13     $S_{\text{SACP}}(x|\hat{y} = l) \leftarrow w_v(x, l) \cdot S_{\text{base}}(x|\hat{y} = l)$      // Score for canonical object

14     $l \in \mathcal{C}(x) \Leftrightarrow S_{\text{SACP}}(x|\hat{y} = l) \leq \tau_\alpha^l$   // Conservative inclusion of $\hat{y} = l$ (Eq. 3)

15   **end**

16   **return** $\mathcal{C}(x)$ *for all* $x \in \mathcal{X}$

---

in which $f_\Theta(x', \hat{y})$ is the outcome of the pretrained segmentation model associated with the label $\hat{y}$ when classifying the voxel $x'$.

We also compute the confidence of segmentation model defined as the predictive probability associated with the canonical object label $l$ (e.g., a tumor) and denoted by $p(\hat{y} = l|x)$ for each voxel $x$. High confidence associated with the canonical object label indicates that the model is making reliable predictions that a voxel belongs to that label, which can be valuable in improving reliability in high-risk tasks. The weight function is formulated to represent lower values with higher probabilities (i.e., more confident predictions) and vice versa, emphasizing regions where the model is more confident while discounting less certain regions. To refine the weight computation, we use this segmentation confidence to calculate

the surprisal function $\mathcal{I}(l) \overset{\text{def}}{=} -\log p(\hat{y} = l|x)$. This surprisal quantifies the information content or unexpectedness of observing the canonical object $l$ with probability $p(\hat{y} = l|x)$ and accounts for the model's inherent uncertainty during segmentation. By incorporating surprisal, the prediction sets dynamically adapt to the probabilistic confidence of the model.

Following class-conditional CP with the desired confidence level $1 - \alpha$ and according to Equation (1), we independently compute the class-specific quantile $\tau_\alpha^{\hat{y}}$ associated with the canonical object label $\hat{y} = l \in \mathcal{Y}$, based on the $S_{\text{base}}$ scores of calibration data. Then, we use $S_{\text{SACP}}$ during testing to include the canonical object label $l$ in the voxels' prediction sets as proposed in Corollary 4.

## Appendix C. Proof of Theorem 3

**Proof** Following class-conditional CP, $\tau_\alpha^{\hat{y}}$ denotes the $(1 - \alpha)$-quantile of $S_{\text{base}}$ scores associated with calibration data with label $\hat{y}$. Then, for each voxel $x$, the condition for inclusion the canonical object label $\hat{y} = l$ in the prediction set $\mathcal{C}_{\text{base}}(x)$ generated by $S_{\text{base}}$ scores is:

$$S_{\text{base}}(x|\hat{y} = l) \leq \tau_\alpha^l \ . \tag{10}$$

By the definition of in Equation (3), $S_{\text{SACP}}$ is computed for each voxel $x$ and the canonical object label $\hat{y} = l$ using the normalized weight $w_v$ as,

$$S_{\text{SACP}}(x|\hat{y} = l) = w_v \cdot S_{\text{base}}(x|\hat{y} = l) \qquad s.t. \qquad w_v = \sigma(\tilde{w}_v) \ , \tag{11}$$

where $\tilde{w}_v \in \mathbb{R}^+$ is the raw weight value defined in Equation (4), and $\sigma(.)$ is the steep sigmoid function (with the gain factor $\beta$) defined as $\sigma(\tilde{w}_v) = \frac{1}{1+\exp(-\beta\tilde{w}_v)}$. For other labels $\hat{y} \neq l$, $S_{\text{base}}$ is used to include the labels in the sets. As $\tilde{w}_v$ is positive and normalized to be less than 1, so $0.5 \leq w_v < 1$. Then, it follows that:

$$\forall x \in \mathcal{X}: \quad S_{\text{SACP}}(x|\hat{y} = l) < S_{\text{base}}(x|\hat{y} = l) \ . \tag{12}$$

Note that $\lim_{\tilde{w}_v \to +\infty} w_v = 1$, and consequently, $\lim_{\tilde{w}_v \to +\infty} S_{\text{SACP}} = S_{\text{base}}$. According to Equations (10) and (12), the above inequality implies the following condition to include $l$ in the set:

$$S_{\text{SACP}}(x|\hat{y} = l) = w_v \cdot S_{\text{base}}(x|\hat{y} = l) \leq \tau_\alpha^l \ . \tag{13}$$

Therefore, any label $\hat{y} \neq l$ included in $\mathcal{C}_{\text{base}}(x)$ (i.e., $S_{\text{base}}(x|\hat{y}) \leq \tau_\alpha^{\hat{y}}$) is also included in the prediction set $\mathcal{C}_{\text{SACP}}(x)$ generated by SACP, and for the canonical object $\hat{y} = l$, $S_{\text{SACP}}(x|\hat{y} = l) < S_{\text{base}}(x|\hat{y} = l)$ holds. Formally, this means:

$$\mathcal{C}_{\text{base}}(x) \subseteq \mathcal{C}_{\text{SACP}}(x) \ . \tag{14}$$

∎

## Appendix D. Conservativeness in Conformal Prediction

Conformal prediction constructs set-valued predictions with a user-specified coverage guarantee, ensuring that the empirical coverage of the prediction sets is at least the nominal confidence level. Given a dataset $\mathcal{D}_n = \{(x_i, y_i)\}_{i=1}^n$ and a new test point $x_{n+1}$, CP produces a prediction set $\mathcal{C}_{n,\alpha}(x_{n+1})$ such that

$$\mathbb{P}(y_{n+1} \in \mathcal{C}_\alpha(x_{n+1})) \geq 1 - \alpha . \tag{15}$$

This property, known as *conservativeness*, guarantees that the probability of the true label being included in the prediction set is at least $1 - \alpha$, often making CP slightly over-conservative due to the discrete nature of rank-based p-values in finite samples.

Conservativeness leads to both lower and upper bounds on the empirical coverage. The lower bound is given directly by the validity guarantee, ensuring Equation (15). However, the actual coverage can be higher than $1 - \alpha$ due to the discreteness of conformity scores, leading to an upper bound of the form

$$\mathbb{P}(y_{n+1} \in \mathcal{C}_\alpha(x_{n+1})) \leq 1 - \alpha + \frac{1}{n+1} . \tag{16}$$

This small excess coverage diminishes as $n$ grows, ensuring that CP becomes *asymptotically exact*, meaning

$$\lim_{n \to +\infty} \mathbb{P}(y_{n+1} \in \mathcal{C}_\alpha(x_{n+1})) = 1 - \alpha . \tag{17}$$

For class-conditional CP, $n$ refers to the number of calibration samples in each class. We encounter stronger conservativeness for rare classes (e.g., tumor label) as classes with small $n$ suffer from higher over-coverage due to the larger impact of discrete rank-based p-values. Due to asymptotic exactness, as $n \to +\infty$, the upper bound tightens, and class-conditional CP approaches exact coverage in Equation (17). Unlike standard CP, class-conditional CP does not enforce a single global coverage level but rather adapts to the structure of the data, ensuring per-class validity.

Thus, conservativeness guarantees *validity* for all sample sizes while maintaining distribution-free coverage guarantees. Class-conditional CP maintains the fundamental conservativeness of standard CP but is more sensitive to class imbalances, making it particularly useful when fairness across classes is a concern.

## Appendix E. Additional Experimental Results for MSK Dataset

**Dataset Characteristics.** We analyze 30 contrast-enhanced computed tomography (CT) scans from the Memorial Sloan Kettering (MSK) Medical Segmentation Decathlon Pancreas dataset (Simpson et al., 2019), comprising portal venous phase CT scans from Memorial Sloan Kettering Cancer Center (New York, USA). Ground truth segmentations were established through expert abdominal radiologist annotations for pancreatic masses (including cysts and tumors), while surrounding anatomical structures were segmented using TotalSegmentator (Wasserthal et al., 2023). These complementary segmentations were integrated using a hierarchical fusion approach that prioritizes radiologists' tumor delineations over automated organ segmentations. This dataset includes a heterogeneous mix of pancreatic

Table 3: Vessel-specific coverage rates at different proximity zones for the MSK dataset. The missing values ("-") indicate no tumor voxels were predicted near the celiac trunk and hepatic artery at these distances, consistent with the MSK dataset's focus on resectable PDAC cases.

| Vessel | ≤2mm | ≤5mm | ≤10mm | ≤20mm | >20mm |
|--------|-------|-------|--------|--------|--------|
| CeTr | - | - | - | 0.985 | 0.980 |
| HA | - | 0.956 | 0.753 | 0.906 | 0.991 |
| SMA | 1.000 | 0.995 | 0.997 | 0.991 | 0.975 |
| PV | 0.821 | 0.858 | 0.918 | 0.967 | 0.991 |
| SMV | 0.973 | 0.989 | 0.986 | 0.988 | 0.966 |

masses including resectable PDAC, intraductal papillary mucinous neoplasms (IPMN), and pancreatic neuroendocrine tumors (PNET). This composition notably differs from both the typical clinical presentation of PDAC, where approximately $80 - 85\%$ of patients present with vessel involvement indicating borderline resectable, locally advanced, or metastatic disease, and from our primary dataset which specifically captured the full range of PDAC presentations including locally advanced cases.

**Coverage Analysis.** Our framework maintains strong performance on the MSK dataset, achieving an overall coverage of 0.980 (mean per-case: $0.985 \pm 0.007$ standard error of the mean (SEM)). The coverage significantly exceeds the target coverage of 0.95 (Wilcoxon signed-rank test, $p = 0.0009$).

**Distance-Based Analysis.** Table 3 presents vessel-specific coverage rates across different proximity zones. The coverage patterns reflect the resectable nature of the cases, with notably high coverage rates in regions farther from vessels. Near-vessel regions ($\leq$ 2mm) show more variable coverage (0.821-1.000) when tumor-vessel contact is present. The relative width ratio (RWR) analysis shows a consistent relationship between prediction set size and vessel proximity, though less pronounced than in the primary dataset. Mean RWR values range from $1.141 \pm 0.031$ SEM in near-vessel regions ($\leq$ 2mm) to $1.655 \pm 0.005$ SEM beyond 20mm. This pattern of increasing width with vessel proximity persists across all vessels.

The results from this dataset complement our primary analysis while highlighting the importance of dataset composition in evaluating conformal prediction frameworks for PDAC segmentation. The predominantly resectable cases in the MSK dataset provide insights into framework performance in scenarios with limited vessel involvement, while underscoring the need for diverse datasets that capture the full spectrum of PDAC presentations for comprehensive validation.

**Comparison with Standard Class-Conditional CP.** As described in Figure 3, our spatially-adaptive approach yields comparable overall coverage (0.980 vs 0.979) while demonstrating improved stability in anatomically critical regions. Near vessels ($\leq$ $2mm$), we achieve higher coverage (0.959 vs 0.956) with more efficient prediction sets (RWR $1.141 \pm 0.061$ SEM vs $1.205 \pm 0.095$ SEM). The framework shows a more controlled increase in RWR with vessel proximity, ranging from $1.141 \pm 0.061$ SEM at $\leq 2mm$ to $1.655 \pm 0.009$ SEM beyond 20mm, demonstrating effective adaptation to anatomical context while maintaining strong coverage guarantees.

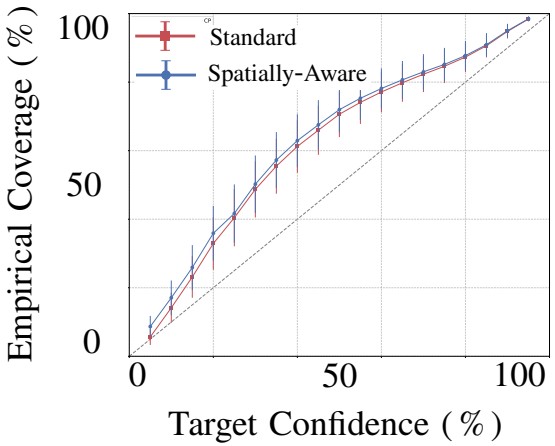

Figure 3: Comparison of empirical coverage at different confidence levels between our method (SACP) and standard Class-Conditional CP (CCCP) on the MSK dataset.

## Appendix F. Experimental Setup Details

### F.1. PDAC Segmentation Model Implementation

The PDAC and organ segmentation model utilized a novel tripartite architecture consisting of a teacher, professor, and student model, implemented using 3D UNet cascade architectures. The teacher model was initially trained on 517 contrast-enhanced CT scans from the PREOPANC trials (Amsterdam UMC and Leiden UMC), LAPC registry (Dutch Pancreatic Cancer Group), and control patients who underwent CT prior to transcatheter aortic valve implantation (Van Tienhoven et al., 2018; Janssen et al., 2021; Stoop et al., 2022). Ground truth segmentations were established by three expert radiologists at the Amsterdam University Medial Centers who manually segmented PDAC tumors in 256 LAP-CTs from 120 patients with (borderline) resectable PDAC and 66 LAP-CTs from 66 LAPC patients using 3D Slicer (version 4.11.20210226 (Fedorov et al., 2012)). Additional anatomical context was provided through automated segmentation of surrounding structures (pancreas, duodenum, spleen, kidneys, adrenal glands, liver, and gallbladder) using TotalSegmentator version 1.5.6 (Wasserthal et al., 2023). The professor model, trained on 106 CT scans, was designed to refine the teacher's pseudo-segmentations using an Underestimation Focuser correction matrix that prioritized correctly identified tumors and areas of underestimation. The final student model was trained on an expanded dataset of 1085 CTs from 903 patients, combining manually segmented data with professor-corrected pseudo-segmentations. The model weights are publicly available at https://zenodo.org/records/14782552.

### F.2. Vessel Segmentation Model Implementation

The vessel segmentation model was implemented using a 3D nnUNet cascade architecture (low-resolution followed by full-resolution) trained on a dataset of 92 contrast-enhanced CT scans (Isensee et al., 2021). The model was designed to segment nine vascular structures: aorta, celiac trunk, hepatic artery, splenic artery, superior mesenteric artery, in-

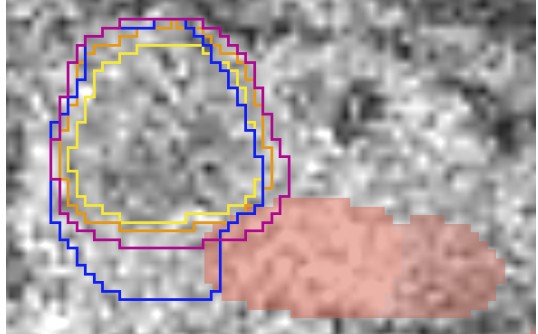

HA contact point: Purple boundary's lateral expansion suggests possible arterial involvement requiring arterial resection planning, while orange CP misses this critical region.

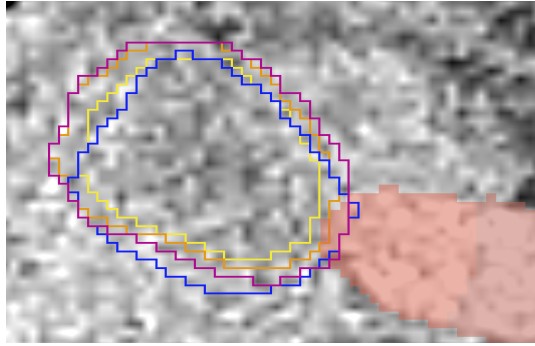

SMA contact point: Spatially-adaptive expansion identifies possible arterial invasion, a distinction missed by uniform CCCP bounds.

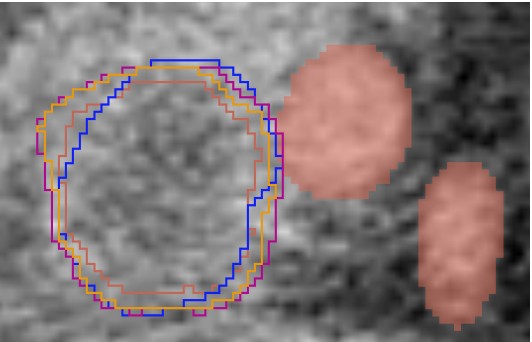

SMV contact point: Purple boundary's circumferential expansion indicates potential venous involvement unlike CCCP's assessment.

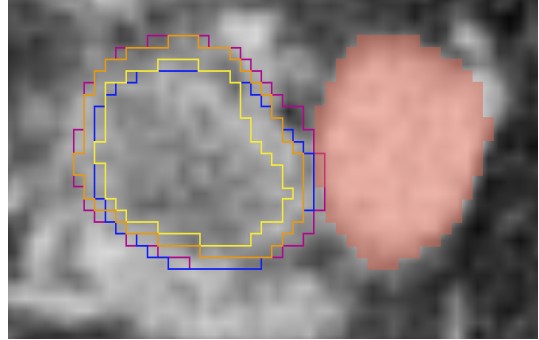

Portal-SMV confluence: Focused purple expansion suggests confluence involvement requiring vascular reconstruction planning, which uniform CCCP bounds fail to detect.

Figure 4: Anatomically-adaptive conformal prediction sets compared to standard CCCP for PDAC cases. Ground truth tumor boundaries (blue), model predictions (yellow), and vessel regions (red) are shown. Our prediction sets (purple) provide adaptive uncertainty bounds based on vessel proximity, unlike the uniform width of standard CCCP (orange), enabling more informed surgical planning in critical regions.

ferior vena cava, portal vein, splenic vein, and superior mesenteric vein. Training data was sourced from the PREOPANC trials and control patients, comprising CT scans from patients with varying stages of pancreatic ductal adenocarcinoma (PDAC) and control subjects who underwent CT imaging for transcatheter aortic valve implantation (Van Tienhoven et al., 2018). Ground truth segmentations were established through manual annotation by seven trained observers at the Amsterdam University Medical Centeres using 3D Slicer (version 4.11.20210226) (Fedorov et al., 2012), with particular focus on the five vessels critical for PDAC resectability assessment: celiac trunk, hepatic artery, por-

Table 4: Dice coefficient analysis by vessel proximity for PDAC segmentation on the PANORAMA dataset.

| Region | Median Dice | Mean Dice | Std Dev |
|---|---|---|---|
| Overall | 0.8084 | 0.7495 | 0.1476 |
| Near vessels (<5mm) | 0.7539 | 0.6364 | 0.2674 |
| Far from vessels (≥5mm) | 0.8161 | 0.7543 | 0.1544 |

tal vein, and the superior mesenteric vessels. The model weights are publicly available at https://zenodo.org/records/14782552.

## Appendix G. Additional Visualization Examples

Figure 4 shows additional examples of our spatially-adaptive conformal prediction method across different PDAC cases taken from the PANORAMA dataset, demonstrating how the prediction sets adapt to varying tumor-vessel relationships. The visualization boundaries are obtained by creating a binary segmentation mask where voxels are assigned a value of 1 if the tumor label is included in their prediction set C(x), and 0 otherwise. The boundary of this binary mask defines our prediction set visualization (shown in purple), while standard CCCP bounds are shown in orange for comparison.

The examples illustrate various clinically relevant scenarios of tumor-vessel interfaces. Near the hepatic artery contact point, the spatially-adaptive boundary expands laterally to indicate possible arterial involvement. At the superior mesenteric artery interface, our method identifies potential arterial invasion through targeted expansion. The superior mesenteric vein contact region shows circumferential expansion suggesting venous involvement, while at the portal-SMV confluence, focused expansion indicates potential involvement requiring vascular reconstruction consideration. These cases demonstrate how SACP provides anatomically-informed uncertainty bounds that adapt based on vessel proximity, offering more detailed information for surgical planning compared to uniform CCCP bounds.

## Appendix H. Segmentation Performance Analysis by Vessel Proximity

To further validate the rationale for spatially-adaptive uncertainty quantification, we analyzed segmentation performance as a function of distance from critical vascular structures across both the PANORAMA and MSK datasets. This analysis revealed consistent patterns of reduced segmentation accuracy near vessel interfaces, indicating that the phenomenon is inherent to the task rather than dataset-specific. Tables 4 and 5 present the Dice coefficient analysis for tumor segmentation stratified by proximity to vessels.

In the PANORAMA dataset, we observed a 12% decrease in mean Dice scores for regions near vessels compared to regions farther from vessels, along with significantly higher variability in near-vessel performance. This pattern was even more pronounced in the MSK dataset, which showed a 38% decrease in mean Dice scores near vessels, with the median score dropping from 0.3553 in distant regions to just 0.0716 near vessels.

Table 5: Dice coefficient analysis by vessel proximity for pancreatic tumor segmentation on the MSK dataset.

| Region | Median Dice | Mean Dice | Std Dev |
|---|---|---|---|
| Overall | 0.3437 | 0.3799 | 0.3078 |
| Near vessels (<5mm) | 0.0716 | 0.2392 | 0.2831 |
| Far from vessels (≥5mm) | 0.3553 | 0.3884 | 0.3138 |

The overall lower Dice scores in the MSK dataset compared to PANORAMA can be attributed to several factors: (1) the MSK dataset includes a more heterogeneous mix of pancreatic pathologies beyond PDAC, including various cystic neoplasms and neuroendocrine tumors that present different imaging characteristics; and (2) annotation protocols between datasets likely differed in how tumor boundaries were defined. Despite these differences in overall performance, the spatial pattern of substantially decreased accuracy near vessels persists across both datasets.

The consistency of this pattern across datasets with different characteristics underscores the fundamental challenge in accurately segmenting tumor-vessel interfaces—precisely the regions where clinical decision-making is most critical for treatment planning. These findings provide evidence for spatially-adaptive uncertainty quantification approaches like SACP, which can account for this predictable spatial heterogeneity by providing appropriately expanded prediction sets in anatomically critical regions.

## Appendix I. Computational Requirements

The computational analysis was conducted on a MacBook Air equipped with an Apple M2 chip and 8GB of RAM, without GPU acceleration. The computational workflow comprised two primary phases: calibration and testing. The pre-computation of probability maps and distance maps is performed separately, with these artifacts serving as input to our conformal prediction framework. The computation of the tumor and vessel distance maps takes approximately 10 and 56 seconds for all 30 scans, respectively and is completed on the cropped scans. The calibration phase, which involves computing non-conformity scores and deriving quantile thresholds across anatomical labels, required approximately 39 seconds. The subsequent testing phase, which generates prediction sets and evaluates uncertainty quantification, took approximately 208 seconds.

## Appendix J. Effect of Vessel Relevancy Hyperparameter ($\gamma$) on SACP Performance

### J.1. Analysis of Different Vessel Weight Configurations

The selection of vessel Relevancy Hyperparameter ($\gamma$) significantly impacts the behavior of the spatially-adaptive conformal prediction framework. As demonstrated in Table 6, we systematically evaluated various configurations to understand their effect on coverage guarantees and prediction set sizes on the PANORAMA dataset.

Table 6: Impact of vessel relevancy hyperparameter ($\gamma$) on performance metrics

| Configuration | Overall Coverage | Coverage ($\leq$2mm) | Coverage (>20mm) | RWR ($\leq$2mm) | RWR (>20mm) | RWR Ratio* |
|---|---|---|---|---|---|---|
| Baseline (arterial: 0.8, venous: 0.6) | 0.989 | 0.984 | 0.988 | 2.739 | 2.525 | 1.085 |
| Min difference (art: 0.8, ven: 0.8) | 0.989 | 0.984 | 0.989 | 2.720 | 2.524 | 1.078 |
| Max difference (art: 0.5, ven: 1.0) | 0.989 | 0.979 | 0.989 | 2.697 | 2.516 | 1.072 |
| CeTr focus ($\gamma$=0.6) | 0.992 | 0.991 | 0.991 | 2.642 | 2.527 | 1.045 |
| HA focus ($\gamma$=0.6) | 0.992 | 0.988 | 0.992 | 2.657 | 2.523 | 1.053 |
| SMA focus ($\gamma$=0.6) | 0.991 | 0.986 | 0.990 | 2.669 | 2.528 | 1.056 |
| PV focus ($\gamma$=0.6) | 0.991 | 0.985 | 0.991 | 2.678 | 2.523 | 1.061 |
| SMV focus ($\gamma$=0.6) | 0.990 | 0.990 | 0.990 | 2.697 | 2.534 | 1.064 |

*RWR Ratio = $\rho(\leq 2mm)/\rho(> 20mm)$

## J.2. Differential Weighting

Our baseline configuration with lower $\gamma$ values for arterial vessels ($\gamma$=0.6) compared to venous vessels ($\gamma$=0.8) maintained strong coverage guarantees (0.989) while providing pronounced spatial adaptation (RWR Ratio: 1.085). This configuration aligns with clinical priorities where arterial involvement typically poses greater surgical challenges.

## J.3. Varying the Contrast Between Vessel Types

Configurations with minimal contrast between arterial and venous weights (art: 0.8, ven: 0.8) and maximized contrast (art: 0.5, ven: 1.0) both demonstrated strong performance. The maximized contrast configuration showed a slight decrease in near-vessel coverage (0.979) compared to other configurations, suggesting a potential coverage-efficiency tradeoff when the contrast becomes too pronounced.

## J.4. Vessel-Specific Configurations

Individual vessel-focused configurations revealed interesting patterns:

1. **Celiac Trunk (CeTr) focus**: Produced the highest near-vessel coverage (0.991) with relatively minimal spatial adaptation (RWR Ratio: 1.045)

2. **Hepatic Artery (HA) focus**: Balanced coverage (0.988 near vessels) with moderate spatial adaptation (RWR Ratio: 1.053)

3. **Superior Mesenteric Artery (SMA) focus**: Similar to HA but with slightly more pronounced spatial adaptation

4. **Portal Vein (PV) focus**: Maintained good coverage with increased spatial adaptation (RWR Ratio: 1.061)

5. **Superior Mesenteric Vein (SMV) focus**: Demonstrated excellent near-vessel coverage (0.990) with strong spatial adaptation (RWR Ratio: 1.064)

Table 7: Coverage and prediction set width comparison across vessel proximity zones

| Distance | CCCP | | UC-CCCP | | SACP | |
|---|---|---|---|---|---|---|
| | Coverage | Width | Coverage | Width | Coverage | Width |
| ≤2mm | $0.954 \pm 0.027$ | $2.887 \pm 0.320$ | $1.000 \pm 0.000$ | $1.608 \pm 0.160$ | $0.981 \pm 0.008$ | $2.762 \pm 0.150$ |
| ≤5mm | $0.970 \pm 0.016$ | $2.702 \pm 0.262$ | $1.000 \pm 0.000$ | $1.691 \pm 0.175$ | $0.987 \pm 0.004$ | $2.684 \pm 0.131$ |
| ≤10mm | $0.977 \pm 0.016$ | $2.611 \pm 0.263$ | $1.000 \pm 0.000$ | $1.758 \pm 0.187$ | $0.988 \pm 0.004$ | $2.621 \pm 0.122$ |
| ≤20mm | $0.978 \pm 0.003$ | $2.574 \pm 0.205$ | $1.000 \pm 0.000$ | $1.835 \pm 0.191$ | $0.987 \pm 0.001$ | $2.592 \pm 0.090$ |
| >20mm | $0.982 \pm 0.002$ | $2.509 \pm 0.078$ | $1.000 \pm 0.000$ | $1.697 \pm 0.020$ | $0.988 \pm 0.000$ | $2.525 \pm 0.036$ |
| Overall | $0.968 \pm 0.038$ | $2.657 \pm 0.226$ | $1.000 \pm 0.000$ | $1.718 \pm 0.147$ | $0.987 \pm 0.004$ | $2.637 \pm 0.106$ |

### J.5. Clinical Implications

The choice of vessel importance weights should align with clinical guidelines and surgical priorities. For centers following NCCN guidelines (United States), where arterial involvement beyond 180° renders a tumor unresectable, configurations with lower $\gamma$ values for arterial vessels (our baseline) may be preferable. For centers following European guidelines with different resectability criteria, alternative weightings may be more appropriate.

Our experiments confirm that the $\gamma$ parameter provides an effective mechanism for tuning the spatial awareness of the conformal prediction framework while maintaining strong coverage guarantees across all configurations.

## Appendix K. Comparison with Uniformly Conservative CCCP

To evaluate whether spatial awareness provides benefits beyond simply increasing overall conservativeness, we compared our SACP approach with a uniformly conservative CCCP (UC-CCCP) on the PANORAMA dataset. The uniformly conservative CCCP applies the minimum weight factor from our spatial approach (0.5) uniformly across all voxels, representing the most conservative setting possible under our weighting scheme.

### K.1. Experimental Setup

We implemented the uniformly conservative CCCP by modifying the non-conformity score as,

$$S_{\text{UC-CCCP}}(x|\hat{y} = l) = 0.5 \cdot S_{\text{base}}(x|\hat{y} = l) \ . \tag{18}$$

This reduces all non-conformity scores for the canonical object by 50%, creating prediction sets that are uniformly more conservative regardless of vessel proximity. We evaluated this approach on the same 20 test cases using identical metrics as our primary experiments.

### K.2. Results

Table 7 shows the comparative results between standard CCCP, uniformly conservative CCCP (UC-CCCP), and our spatially-adaptive approach (SACP).

The results reveal a critical limitation of the uniformly conservative approach: UC-CCCP achieves perfect coverage (1.000) across all regions, but this comes with a fundamental change in the prediction set structure. The perfect coverage indicates that for every

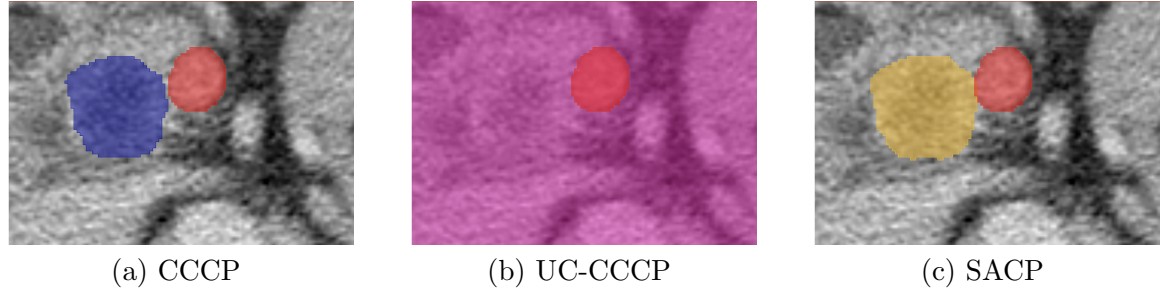

| (a) CCCP | (b) UC-CCCP | (c) SACP |

Figure 5: Comparison of prediction sets across different conformal prediction methods. Blue shows standard CCCP prediction sets, pink represents uniformly conservative CCCP sets, orange illustrates SACP prediction sets, and red represents the Superior Mesenteric Vein. The figure demonstrates how SACP provides more nuanced uncertainty quantification compared to standard and uniformly conservative CCCP approaches.

voxel, the prediction set includes all possible labels - effectively rendering the predictions meaningless from a clinical perspective.

The reported width values for UC-CCCP might appear smaller than CCCP and SACP, but this is misleading because the width metric measures average cardinality relative to the total number of classes. When a prediction set includes all possible labels (as happens with UC-CCCP), the prediction set width calculation is affected by how the metric is normalized. The actual prediction sets are maximally large, including all possible label options for every voxel, which renders them clinically unusable.

In contrast, SACP maintains high coverage (0.987) while producing prediction sets that spatially adapt to anatomical context. As shown in Figure 5, SACP creates tighter prediction sets away from vessels and more conservative sets near vessels, aligning with clinical priorities for surgical planning. Additional image comparisons are presented on our github https://github.com/tailabTMU/SACP.

These results demonstrate a fundamental advantage of SACP: it achieves the necessary balance between coverage guarantees and prediction set efficiency. Simply making CCCP uniformly more conservative leads to perfect but clinically useless prediction sets. SACP, on the other hand, focuses conservativeness where it matters most - near vessels where accurate boundary delineation is crucial for surgical planning - while maintaining efficient prediction sets elsewhere.

## Appendix L. Analysis of Binary versus Continuous Weighting Schemes

The binary weighting approach represents a simplified baseline for incorporating anatomical context into conformal prediction. To ensure fair comparison, we derived the binary weights from our continuous method, using 0.5 and 0.997 as they represent the minimum weights applied across all voxels and cases for near ($< 5$mm) and far ($> 5$mm) regions respectively. This choice ensures the binary method maintains at least the same level of conservativeness as our continuous approach in each region. This method divides the prediction space into two distinct regions based on proximity to critical structures:

$$w_{\text{binary}}(x) = \begin{cases} w_{\text{near}} & \text{if } \delta_v \leq d_{\text{threshold}} \\ w_{\text{far}} & \text{if } \delta_v > d_{\text{threshold}} \end{cases} \qquad (19)$$

where $\delta_v$ represents the distance to the nearest vessel $v$, $d_{\text{threshold}}$ is a fixed distance threshold (e.g., 5mm), and $w_{\text{near}}$, $w_{\text{far}}$ are predetermined weights for near and far regions, respectively.

To compare this baseline against our spatially-adaptive approach, we implemented the binary scheme using weights of 0.5 and 0.997 for near ($< 5$mm) and far ($> 5$mm) regions respectively, derived from the empirical weight distribution of our original method. The binary approach achieved the following results:

- Strong nominal coverage: Vessel-wise coverage reached 1.000 for regions within 5mm across all vessels

- Maintained coverage of 0.985-0.989 for regions beyond 10mm

- Prediction sets exhibited significantly larger volumes near vessels compared to our continuous approach

However, this apparently strong performance revealed several limitations:

1. **Spatial Discontinuity**: The sharp transition at the 5mm boundary creates artificial discontinuities in prediction sets that do not reflect the gradual nature of anatomical relationships

2. **Over-conservative Estimates**: The binary approach tends to include entire vessel-adjacent regions in prediction sets, leading to unnecessarily large prediction regions

3. **Loss of Anatomical Context**: The simplified weighting scheme fails to capture the nuanced spatial relationships present in medical images

The limitations of the binary approach become immediately apparent in Figure 6. While achieving strong numerical coverage ($> 0.98$ across all vessel-wise evaluations), the binary method produces prediction sets that are too broad to be clinically useful. As shown in Figure 6($b$), it effectively includes entire vessel-adjacent regions in its prediction sets. Additional image comparisons are presented on our github https://github.com/tailabTMU/SACP. In contrast, our continuous weighting approach offers several advantages:

- Smooth transitions in prediction set boundaries

- Adaptive uncertainty estimation based on continuous distance measures

- Integration of multiple anatomical factors through $\phi_l$ and $\mathcal{I}(l)$

## Appendix M. Additional Analysis of Relative Width Ratio

The Relative Width Ratio (RWR) provides a quantitative measure of how prediction set sizes adapt based on proximity to critical anatomical structures. Table 8 presents a comprehensive analysis of both coverage and RWR values across different vessels and distance thresholds. The RWR values demonstrate several key patterns:

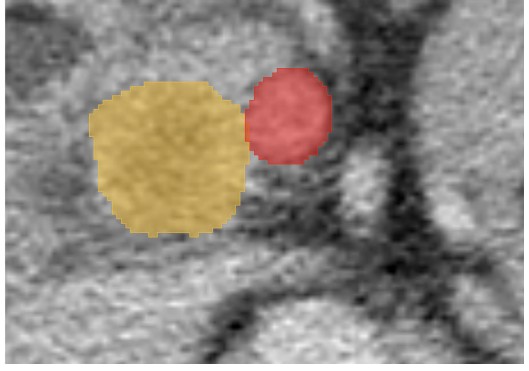
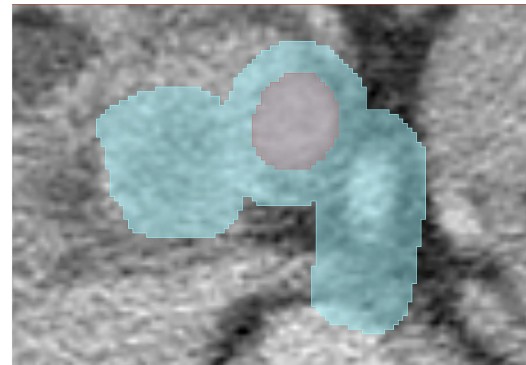

(*a*) SACP (Our Method)         (*b*) Binary Threshold Baseline

Figure 6: Visual comparison between our spatially-adaptive approach and the binary threshold baseline. Both images show a pancreatic tumor (yellow) adjacent to a critical vessel (red). While both methods achieve the target coverage rate, the binary approach (b) produces clearly unsuitable prediction sets. It effectively includes almost the entire vessel-adjacent region (light blue), failing to provide meaningful uncertainty bounds for surgical planning. In contrast, our SACP method (a) generates focused prediction regions (orange) that maintain anatomical relevance while ensuring coverage.

Table 8: Vessel-specific coverage rates and Relative Width Ratios (RWR) at different proximity zones

| Vessel | 2mm | | | | 5mm | | | | 10mm | | | | 20mm | | | | >20mm | | | |
|---|---|---|---|---|---|---|---|---|---|---|---|---|---|---|---|---|---|---|---|---|
| | CCCP | | SACP | | CCCP | | SACP | | CCCP | | SACP | | CCCP | | SACP | | CCCP | | SACP | |
| | Cov | RWR | Cov | RWR | Cov | RWR | Cov | RWR | Cov | RWR | Cov | RWR | Cov | RWR | Cov | RWR | Cov | RWR | Cov | RWR |
| CeTr | 0.999 | 2.539 | 1.000 | 2.569 | 0.999 | 2.570 | 1.000 | 2.563 | 0.998 | 2.650 | 1.000 | 2.630 | 0.980 | 2.720 | 0.987 | 2.722 | 0.980 | 2.455 | 0.988 | 2.471 |
| HA | 0.959 | 3.369 | 0.980 | 3.275 | 0.973 | 3.136 | 0.986 | 3.153 | 0.987 | 3.007 | 0.994 | 3.040 | 0.981 | 2.826 | 0.989 | 2.849 | 0.980 | 2.443 | 0.987 | 2.454 |
| SMA | 0.925 | 2.761 | 0.975 | 2.627 | 0.967 | 2.569 | 0.989 | 2.501 | 0.982 | 2.501 | 0.994 | 2.468 | 0.973 | 2.476 | 0.985 | 2.479 | 0.984 | 2.524 | 0.989 | 2.552 |
| PV | 0.927 | 3.152 | 0.953 | 3.006 | 0.955 | 2.847 | 0.974 | 2.877 | 0.957 | 2.698 | 0.974 | 2.743 | 0.978 | 2.614 | 0.989 | 2.642 | 0.981 | 2.468 | 0.987 | 2.478 |
| SMV | 0.960 | 2.615 | 0.997 | 2.331 | 0.956 | 2.387 | 0.987 | 2.327 | 0.958 | 2.201 | 0.980 | 2.222 | 0.975 | 2.236 | 0.987 | 2.270 | 0.983 | 2.655 | 0.987 | 2.672 |

1. **Distance-dependent Adaptation:** Both CCCP and SACP show decreasing RWR values as distance from vessels increases, indicating more precise prediction sets in non-critical regions.

2. **Vessel-specific Behavior:** Arterial vessels (HA, SMA) show higher RWR values in close proximity ($\leq$2mm) compared to venous vessels (PV, SMV), reflecting the clinical importance of arterial involvement in surgical planning.

3. **SACP Improvements:** Our method generally maintains or reduces RWR values while achieving higher coverage, particularly in critical regions ($\leq$5mm from vessels).

4. **Stability at Distance:** Beyond 20mm, RWR values stabilize around 2.4-2.6 for both methods, indicating consistent behavior in non-critical regions.

