# OpenReview forum: "SACP: Spatially-Adaptive Conformal Prediction in Uncertainty Quantification of Medical Image Segmentation"
_MIDL.io/2025/Conference — MIDL 2025 Poster_

### Official Review · Reviewer_R3Em · 2025-02-21

**Confidence:** 4
**Preliminary Rating:** 3
**Recommendation:** Poster
**Final Rating:** 4

**Summary:**

The paper provides a novel adaptation of the classical conformal prediction algorithm for segmentations in order to produce more conservative uncertainties in sub-regions of the image to segment. This approach adapts uncertainty estimation for high-risk regions in images.
The reviewer appreciates the authors' novelty of the work. Still, the paper lacks clarity at some points and some formulations might be misleading. The evaluation and results are not too convincing and the resulting guarantees from just the paper (without considering the appendix) not too clear. The conclusion part lacks showing the limitations of the proposed method (like segmentations of critical masses needed for generating prediction sets).

**Strengths:**

Adaptation of uncertainty quantification in high-risk regions:
This method shows a novel way of incorporating spatially-dependent demands on uncertainty quantification into a calibration algorithm that comes with mathematical guarantees.

**Weaknesses:**

- please note that the reviewer might be wrong in some aspects and is open to clarifications in case of misunderstandings
- major
    - In the third section of the introduction the authors use confusing statements. It is mentioned that one wants higher precision with lower uncertainty in critical areas but as to my understanding, the authors' algorithm is not able to fix this problem but rather produce more conservative (and therefore usually larger) prediction sets in critical areas. A higher precision with lower uncertainty can only be produced by increasing the segmentation performance
    - When surgeons see the final model uncertainty output, how is it indicated for the clinician which region has been weighted how? This should be a crucial point as the weighing might be off and therefore the generated uncertainties are incorrect
    - The algorithm only works under the assumption that the segmentation of the critical masses is correct which is highly unlikely when those are automatically produced by another segmentation network
    - The related work section neglects a paper that was presented at last year's MICCAI conference that uses region-based distribution-free uncertainty quantification with statistical guarantees: Subgroup-Specific Risk-Controlled Dose Estimation in Radiotherapy by Fischer et al.
    - In 2.2 it says that for standard CP the prediction sets are overly conservative for rare classes when using one alpha across all classes. This statement usually doesn't hold true since the calibration is dominated by the largest-occurring class and dependent on this the prediction sets for the minority class can be either overly conservative or (usually) under-conservative.
    - In the experiments and section it would have been more fair to compare the SACP with a CCCP where alpha matches the most conservative setting in the weighted setting (e.g. 5%). This would demonstrate whether it is really useful to use the spatially aware algorithm instead of just making the original CCCP more conservative.
    - the paper works only with the appendix

**Detailed Comments:**

- minor
    - affiliations and names: first all names and then all affiliations; see previous MIDL submissions for examples
    - it should be "non-conformity" instead of "nonconformity"

**Justification Of The Final Rating:**

The paper improved a lot with the new manuscript and I acknowledge that. Therefore I increased my rating to weak accept. However, I do not agree with the authors about their clarification on the automatically generated segmentations for critical structures. A crucial point was even mentioned: even annotations by expert annotators show high variability. That's exactly my point as such ambiguities influence the performance of the authors' proposed algorithm drastically. This uncertainty and the one you model interact with each other. Just to be clear: I am not saying that this is a major flaw of your *method* but it definitely is for your experiments. And I also don't think this makes the paper less publication worthy. It's actually the opposite, I do think that the method holds great value for the community. But the limitations are definitely something that need to be discussed more openly in the paper and I would still love to see something to that in the final version in case of acceptance.

**Justification Of The Preliminary Rating:**

The contribution seems very important and might be very useful for many cases. However, the paper seems to be a bit confusing in some parts and the evaluation against CCCP seems a bit unfair. Also the effect size of the improvement is not that big, even though it is significantly better.
Also, the paper needs to address some crucial limitations more.

**Questions To Address In The Rebuttal:**

1. Clarification of how you ensure the quality/correctness of the automatically generated segmentations of critical masses
2. Clarification of the confusing/misleading sentences mentioned above
3. A more fair comparison, i.e. SACP vs a more conservative $\alpha$ level in the CCCP
4. Include the kind of guarantees and guarantee changes (letting the proof for them in the appendix is fine) in the main article
5. Address the limitations of the method more in the discussion section

---

> ### Author Response · Authors · 2025-03-08
> **Response to Reviewer R3Em**
>
> ***Questions***
>
> **Q1:** Our approach acknowledges the inherent challenges in tumor boundary delineation that exist regardless of model performance.
> However, we emphasize that SACP's fundamental contribution addresses a more nuanced challenge than merely improving segmentation accuracy. The uncertainty at tumor edges represents not just model limitations but genuine clinical ambiguity—even expert radiologists exhibit significant inter-observer variability when delineating tumor boundaries near vessels. Our method is designed to acknowledge this inherent uncertainty rather than attempting to eliminate it through improved segmentation alone.
> SACP functions complementarily to any improvements in segmentation accuracy, adapting its uncertainty estimates based on both model confidence and anatomical context. As such, it acknowledges that perfect segmentation of critical masses is neither possible nor necessary for effective uncertainty quantification in surgical planning.
> In terms of the segmentation quality in our experimental validation, for vessel segmentations (critical masses), we employ a specialized vessel-specific model trained on 92 contrast-enhanced CT scans from the PREOPANC trials. The quality of these vessel segmentations was ensured through:
> - Training on high-quality manual annotations by seven trained observers using 3D Slicer, with particular focus on the five vessels critical for PDAC resectability assessment: celiac trunk, hepatic artery, portal vein, and superior mesenteric vessels.
>  - Leveraging the natural advantages of contrast-enhanced CT imaging, where vessels appear distinctly hyperattenuating compared to surrounding tissues, providing excellent contrast for segmentation.
>  - Implementing a 3D nnUNet cascade architecture (low-resolution followed by full-resolution) that has demonstrated state-of-the-art performance in vascular segmentation tasks.
>
> **Q2:** Addressed in Comment **C1**.
>
> **Q3:** Addressed in Comment **C6**.
>
> **Q4:** We've added a concise statement of theoretical guarantees to Section 2.2 **Spatially-Adaptive Non-Conformity Score** (Page 5), highlighting that SACP maintains the ($1-\alpha$) coverage guarantee while providing spatial adaptation.
>
> **Q5:** We've added a limitations subsection to the Conclusion (Section 4, Page 9) discussing key constraints and future research directions.
>
> ***Comments***
>
> **C1:** This was an oversight and it's now fixed in the Introduction (Page 2) color-coded in red.
>
> **C2:** How to interface SACP with the clinician is an important research topic of Human-Computer Interaction, which our study was limited to address it. We added this limitation and shared the following envision in the SACP Limitations of the manuscript (Section 4, Page 9). One way to design the interface, is a multi-layered visualization approach: (1) solid mask showing base segmentation; (2) semi-transparent overlay indicating prediction set boundaries; (3) color-gradient visualization reflecting confidence levels. This displays the practical impact of spatial weighting without exposing mathematical parameters directly.
>
> **C3:** Our approach is robust to minor segmentation imprecisions through several design elements in Section 2.2 (Page 5): (1) Contrast-enhanced CT provides inherently high vessel contrast; (2) Our continuous distance-based weighting function creates gradual importance decay that accommodates boundary variations; (3) The probability-based component of our weighting function provides additional protection against potential segmentation inaccuracies; (4) The conservative prediction set expansion near perceived vessels acts as a safeguard. Experimental results validate this robustness across different vessel types with varying segmentation challenges.
>
> **C4:** We've included Fischer et al.'s work on subgroup-specific risk-controlled dose estimation in Section 1, Related Work (Page 3) and discussed its relationship to our approach.
>
> **C5:** We've revised our description in Section 2.2 **Spatially-Adaptive Non-Conformity Score** (Page 4) to more accurately reflect class-imbalanced behavior.
>
> **C6:** We implemented a uniformly conservative CCCP (UC-CCCP) by applying our minimum weight factor (0.5) across all voxels. Results in Appendix K (Page 23) show UC-CCCP achieves perfect coverage (1.000) but generates impractically large prediction sets that include almost all labels. SACP maintains high coverage (0.987) while producing significantly more efficient, clinically useful prediction sets. This demonstrates that spatial adaptation provides benefits beyond simply increasing global conservativeness.
>
> **C7:** We've corrected the affiliations format and standardized terminology to "non-conformity" throughout the manuscript.

---

### Official Review · Reviewer_UeC5 · 2025-02-22

**Confidence:** 4
**Preliminary Rating:** 3
**Recommendation:** Poster
**Final Rating:** 4

**Summary:**

The authors explore conformal prediction for medical image segmentation. Concretely, the authors aim to provide better coverage in regions near critical structures, e.g., arteries, when segmenting target organs/tumors. For that purpose, they design a spatial non-conformity score to produce conservative sets near critical structures by applying a penalty on the non-conformity score based on a distance measure. The proposed pipeline is evaluated on a single dataset with a positive outcome.

**Strengths:**

- Conformal prediction (CP) is an increasingly popular framework for reliability assessment. Given its current rise in vision applications, this work timely applies CP to segmentation
- Very few works assess CP for segmentation, and even fewer for medical image segmentation; thus, the application is novel.
- The proposed non-conformity measure is well-motivated, and the results have been successful.

**Weaknesses:**

**[Case-specificity]** The presented non-conformity measure is too specific. It assumes the presence of annotated/segmented critical regions, and its application is extremely tailored toward surgical planning. Its applicability to more general and extended segmentation applications is unclear, which limits the proposed method's impact. From an experimental perspective, it is worth noting that it is only validated in a single dataset.

**[Base non-conformity score and Baselines]** I could not find which base non-conformity measure the authors use - although it appears to be [LAC]. However, there exist other non-conformity measures [APS,RAPS] which provide better adaptability. It would be beneficial if the authors showed the robustness of their method to different base non-conformity scores.

**[Naive baselines]** A simple baseline could be created using only the base non-conformity score and a region-wise threshold. Two thresholds per class would be calculated: one for the inner regions of the object and the other for the critical regions, which can be defined by specific distance thresholds to critical structures, such as the distances in Table 1. Thus, coverage in both areas of interest should be controlled. How would this straightforward baseline compares with the proposed spatially-aware non-conformity score?

**[Correctness of technical claims]**

- “Conformal prediction… fails to account for spatially varying importance…”. CP is a framework, it would be the current non-conformity measures the ones that fail to account for such priors.
- “We develop… enabling robust performance across different imaging protocols and populations”. First, the proposed method is validated on a single dataset. Second, multi-center generalization deployment might not hold the exchangability assumptions if calibration data from each center is not leveraged.
- “To consider uncertainty in the prediction, one may apply CP … to ensure at least 90% certainty around the delineated boundaries”. However, CP provides marginal theoretical coverage guarantees, which means CP does not necessarily allow one to ensure conditional coverage for some pixel regions.
- “CP uses a single threshold alpha across all classes.” This is not necessarily true. Indeed, there are works that have already proposed class-specific quantile search [LAC]. In this line, the authors point out Angelopoulos et al. 2023 for class-conditional conformal predictors. However, the authors of [LAC] introduced it previously (it is the latest reference I know, to my knowledge).

**[Hyper-parameters]** Studies on how the authors have set gamma hyper-parameters  (compared to fixed values across critical structures) and why beta is necessary would provide a deeper understanding of the non-conformity measure.

**[Excheangability in medical image analysis]** The results showcase that the marginal coverage in calibration and test data differ. In this case, the coverage of test data is indeed better. However, such results question the exchangability of such data. This issue is especially challenging in medical image analysis since new patients are expected in test data, and hence, it remains unclear whether such data is exchangeable. The authors should further discuss such issues. Even though the test data coverage improves for the dataset employed,  is there any guarantee that this will hold for other datasets?

**References:**

[LAC]  Least ambiguous set-valued classifiers with bounded error level, Journal of the American Statistical Association, 2019.

[APS] Classification with valid and adaptive coverage, NeurIPS, 2020.

[RAPS] Uncertainty sets for image classifiers using conformal prediction, ICLR, 2020.

**Detailed Comments:**

- The authors should define how the boundaries are obtained from prediction sets for the qualitative visualizations (I could not find it). If I understand properly, a pixel is considered part of the object if the positive category is within the pixel’s set.

**Justification Of The Final Rating:**

The authors have adequately addressed most of the doubts/concerns highlighted by the reviewers. They have incorporated additional ablation studies to the gamma and beta values, baselines, and experiments using an additional dataset. They have also notably improved the clarity of the manuscript and the correctness of technical claims. This work is an interesting development of non-conformity scores in conformal prediction to address challenges in medical image segmentation. Hence, I increase my score.

**Justification Of The Preliminary Rating:**

My main concerns are with the very narrow scenario contemplated in this work [Case-specificity]. The framework's usability in more general segmentation scenarios is unclear, which might limit its impact. Also, I have some concerns regarding the base non-conformity scores used, which do not incorporate adaptability measures [Base non-conformity score and Baselines], and simpler baselines [Naive baselines]. These issues prevent me from providing a positive score, although I have an overall positive view of the work. Least importantly, it would be nice if the authors could elaborate on [Correctness of technical claims], [Hyper-parameters], and [Excheangability in medical image analysis].

**Questions To Address In The Rebuttal:**

See Justification.

**Special Issue:**

No

---

> ### Author Response · Authors · 2025-03-08
> **Response to Reviewer UeC5 (Part 1)**
>
> **C1:** The visualization boundaries are obtained by creating a binary segmentation mask where voxels are assigned a value of 1 if the tumor label is included in their prediction set C(x), and 0 otherwise. In Figure 4, the boundary of this binary mask defines our prediction set visualization (shown in purple), while standard CCCP bounds are shown in orange for comparison. We have now clarified this in Appendix G (Page 21).
>
> **C2:**  While our method was motivated by pancreatic cancer surgical planning, its principles apply to various domains requiring spatially-adaptive uncertainty quantification. Examples include: (1) autonomous driving, where pedestrians and cyclists require different uncertainty thresholds than road surfaces; (2) industrial quality control, where defect detection near crucial components needs higher precision; and (3) natural disaster prediction, where uncertainty in flood boundaries near populated areas carries greater consequences. In all cases, the framework requires a spatially structured prediction task, canonical objects, and critical structures. We've added descriptions of these generalized scenarios (Section 4, **SACP Generalizability Beyond Medical Imaging**, Page 9). We also clarify that our validation spanned two distinct datasets: PANORAMA (30 scans) and MSK (30 scans), with consistent performance across different patient populations and tumor types.
>
> **C3:**  Our implementation indeed uses LAC as the base non-conformity measure, which we have now explicitly stated in Section 2.2. While evaluating multiple base scores would be valuable, SACP's mathematical formulation is designed to be agnostic to the specific base non-conformity measure. Our spatial weighting mechanism applies a multiplicative factor to the base score, preserving the original CP's coverage guarantees regardless of which base measure is used. Since the spatial weighting component operates independently from the base score calculation, we expect similar relative improvements when using APS or RAPS as alternatives. We have clarified this in the manuscript and acknowledge this limitation in our discussion. Exploring SACP with different base non-conformity measures would be interesting future work.
>
> **C4:**  We implemented a binary threshold baseline using two distinct weights (0.5 for regions within 5mm of vessels, 0.997 for regions beyond), chosen as the minimum weights applied by our continuous method across voxels. While achieving strong coverage metrics, visual analysis (Appendix L, Figure 8) reveals that this approach includes entire vessel-adjacent regions in prediction sets, making them clinically unsuitable. This demonstrates why a continuous weighting scheme is necessary for meaningful uncertainty quantification in medical image segmentation.
>
> **C5:**  We have revised our manuscript to clarify that the limitation lies in current non-conformity measures rather than the CP framework itself (Abstract, Page 1).
>
> **C6:**  We addressed multi-center concerns through two approaches: (1) The PANORAMA dataset itself comprises CT scans from five European centers, providing inherent multi-center validation; (2) We extended validation to the MSK dataset, representing different patient populations and more heterogeneous pathologies including resectable PDAC, IPMN, and PNET (Appendix F, Page 18). This out-of-distribution testing demonstrates our method's adaptability beyond a single disease presentation or imaging protocol.
>
> **C7:** We have revised the manuscript to be more accurate in our description of CP's guarantees (Section 2, **Conformal Prediction**, Page 3).
>
> **C8:**  We've clarified that our statement referred to vanilla CP that calibrates a single quantile over all classes in Section 2.2 (Page 4). Our approach exploits class-conditional CP to account for imbalanced distribution in medical imaging, where canonical objects (tumors) are rare compared to other classes. We've revised the manuscript and included the original CCCP citations (Section 2.2, Page 4): Shi et al. (2013) and Sadinle et al. (2019) and SACP generalizability beyond medical imaging.
>
> **C9:**  Appendix J (Page 21) now includes comprehensive analysis of different $\gamma$ configurations. While all maintain strong coverage (0.989-0.992), they differ in how they adapt prediction sets near critical structures. The RWR ratio ranges from 1.045 (CeTr focus) to 1.085 (baseline), showing that $\gamma$ effectively controls spatial adaptation without compromising coverage. Regarding $\beta$, our experiments confirm its necessity for creating sharper transitions in uncertainty estimates (described in Section 3.2, Page 8). Without this parameter, transitions between high/low uncertainty regions become too gradual for clinical utility. Our baseline configuration (arterial: $\gamma=0.6$, venous: $\gamma=0.8$) reflects NCCN guidelines where arterial involvement determines resectability.

---

> > ### Comment · Reviewer_UeC5 · 2025-03-13
> >
> > I would like to thank the authors for their efforts to update the manuscript and address in detail the raised concerns. The authors have clarified my doubts/questions properly, and I agree with reviewer Pciq that the current version has significantly improved the article. Therefore, I will increase my score, as I believe this paper is an interesting contribution to MILD.

---

> ### Author Response · Authors · 2025-03-08
> **Response to Reviewer UeC5 (Part 2)**
>
> **C10:**  The observed higher coverage (0.987 vs 0.95) is expected in conformal prediction due to its inherent conservativeness with discrete quantiles. We addressed exchangeability concerns through careful patient-level partitioning between calibration and test sets. Our validation across two distinct datasets (PANORAMA and MSK) with different populations and protocols demonstrated consistent performance (coverage of 0.987 and 0.980 respectively), suggesting robustness to moderate distribution shifts. While no method guarantees identical performance across arbitrary datasets, SACP preserves conformal prediction's theoretical guarantees while adding anatomical context. For deployment in significantly different settings, we recommend a small local calibration phase (10-15 cases).

---

### Official Review · Reviewer_Pciq · 2025-02-25

**Confidence:** 4
**Preliminary Rating:** 3
**Recommendation:** Poster
**Final Rating:** 4

**Summary:**

This paper introduces SACP (Spatially-Aware Conformal Prediction), a novel framework that extends conformal prediction to incorporate varying spatial importance of voxels in medical image segmentation.

Standard CP provides statistical coverage guarantees (e.g., ensuring a prediction set contains the true label with a specified probability) but treats all regions of an image equally and assumes uniform uncertainty across prediction regions. SACP specifically addresses the challenges in medical image segmentation where different spatial regions - particularly near critical vascular structures - demand distinct levels of certainty measurements. SACP modifies CP to be sensitive to the proximity of voxels to critical and canonical structures. It does so by introducing locally adaptive nonconformity scores using a spatially-aware weight that explicitly accounts for distance to critical structures, enabling CP to generate prediction sets with higher precision (thus lower uncertainty) near critical boundaries while remaining CP-level uncertain elsewhere

In this work, SACP is validated on PDAC tumor segmentation from CT scans, where accurate delineation near vessels is crucial for surgical planning. The method is compared to standard Class-Conditional Conformal Prediction (CCCP).

**Strengths:**

* **Clear Motivation and Explanation**: The paper clearly defines the problem-statement and provides a well-explained solution. The clinical context is well-described. Overall, the paper is very well-written and theoretically sound.

* **Novelty and Strong Methodology**: The core idea of spatially-aware conformal prediction is novel and directly addresses a significant limitation of standard CP in medical image segmentation. The methodology is explained in great detail and the different proofs provided covers a lot of important concepts.

* **Clinical relevance**: The application to PDAC surgical planning is highly clinically relevant.

* **Use of relevant metrics for evaluation**: The evaluation includes both quantitative metrics (coverage, RWR) and qualitative visualizations. Ref. Table 1: SACP achieves higher coverage and more efficient prediction sets especially in near-vessel regions.

* **Reproducibility**: Code has been provided to reproduce the experiments.

**Weaknesses:**

While the methodology is detailed and thoroughly fleshed out, a lot of the weaknesses in this paper also stem from the space given to the various proofs (which could have been easily moved to the supplementary section).  There are glaring gaps in terms of validating the performance, robustness and generalizability of the proposed method, comparing with other models and also discussing the computational costs. More specifically:

* **Accuracy of segmentation** : Accuracy of the AI-generated segmentations has not been mentioned at all. Would the performance of the model affect the computations done using SACP ? If a class-imbalance is present in the dataset, how does this affect the output of SACP ?

* **Complexity of the Weight Function and impact of the different parameters**: The weight function (Equation 4) is relatively complex, involving multiple terms and parameters. While each term is motivated, the overall complexity could make it difficult to interpret or tune in practice.

 For **hyperparameter γ**, while it is mentioned that the NCCN resectability criteria is used for determining its value, it would have been helpful to know how varying γ would affect the coverage.

* **Limited Comparison to Other Uncertainty Methods**: The comparison is primarily to standard CCCP. While SACP is a modification of CP, comparing it to other uncertainty quantification methods (e.g., SACP by Liu et al. (https://www.arxiv.org/pdf/2409.01236) , ensemble methods) would provide a broader context.

* **Computational Cost**: The paper doesn't explicitly discuss the computational cost of SACP compared to standard CCCP. There is no mention of the training process and the hardware used. While the cropping strategy aims to improve efficiency, computing the distance maps and the spatially-aware weights likely adds some overhead. A few sentences about these would have been useful.

* **Baseline method** : SACP is evaluated against the CCCP - however there seems to be no ground truth available to compare the distribution and accuracy of the coverage and RWR values.

* **Variable performance improvement reported in Table 2** : The results for CeTr show minimal improvement across the distances. For other vessels, there is a consistently significant improvement at a distance of < 2mm but variable improvements in all the other distance ranges.

* **Limited Validation**: Given the strong underlying methodology, it would have been great to see this validated on at least one more dataset. Even in the validation done in this work using the publicly available PANORAMA dataset, the validation set size is small (n=30) compared to the number of publicly available scans.: https://panorama.grand-challenge.org/datasets-imaging-labels/

* **Missing references to recent papers** : A few recent papers focusing on local spatial conformal prediction have not been cited. Please refer to the rebuttal section for a list of these papers.

**Detailed Comments:**

-------------------
**Minor Corrections**

* Empirical coverage rate (eqn .6) : define this more thoroughly. While the equation itself is explained, please expand upon the definition of coverage in terms of clinical relevance.

* AI-Generated Segmentations: provide a sentence or two on the dataset used. Looking at the reference provided (Bereska et al. 2024), this is a teacher-professor-student model implemented using a 3D U-net. Please explicitly mention this along with the dataset.

* If accepted to the main conference: include a line or two from Appendix D defining conservativeness.
--------------------

**Justification Of The Final Rating:**

The paper proposes a novel method SACP that addresses the limitations of standard Conformal Prediction (CP) in scenarios where different spatial regions require distinct certainty measurements. This can have a strong clinical impact especially in surgical interventions where the operability of a tumor is determined by the certainty in which the tumor and surrounding structures are detected.

The methodology outlined in this paper is very rigorous and backed with sound theoretical proofs. The revised version submitted by the authors after the rebuttal is more organized and addresses a lot of questions raised by the reviewers. Some weaknesses are evident: using a small subset of the large public dataset "PANORAMA" for validating the method; having lot of redundant information in the methodology section. However, the merits of this paper outweigh these weaknesses.

I am keen on seeing this paper in the MIDL conference. It will be interesting to see how this work evolves. This has the potential to be a strong and relevant paper for the medical imaging community and may also invite interest from researchers in other domains as well.

**Justification Of The Preliminary Rating:**

The paper proposes a novel method SACP that addresses the limitations of standard Conformal Prediction (CP) in scenarios where different spatial regions require distinct certainty measurements. This can have a strong clinical impact especially in surgical interventions where the operability of a tumor is determined by the certainty in which the tumor and surrounding structures are detected.

The methodology outlined in this paper is very rigorous and backed with sound theoretical proofs. However, in context of the MIDL conference, there is very little space given to explaining the training experiments and validation of this method. A very small subset of the large public dataset "PANORAMA" has been used for validating the method which is insufficient. Some of the metrics required for analysis are missing and the subsequent improvements over CCCP especially those reported in Table 2 are highly variable.

This has the potential to be a strong and relevant paper if the contents are restructured and more experiments are done to support the methodological claims. A lot of details are given in the supplementary sections (especially in Appendices B-D) which would have been helpful in the main paper, and vice-versa.

**Questions To Address In The Rebuttal:**

* **More Validation needed**: Please provide more details on the 30 scans selected for the validation study (were these selected at random or using a pre-defined criteria) ? And why was only a small subset used from the larger PANORAMA dataset ? This would help in addressing the presence of class imbalances or bias (if any) in the dataset and the robustness of SACP.

--------

* **Metrics for assessment**:

 -- Include the size-stratified Coverage Violation (SSCV) to look at how the coverage rate varies depending on the size of the prediction sets. This will help in detecting potential issues with the calibration of the proposed conformal prediction method (SACP)

-- In Table 2, please report the RWR scores as well

-- Please mention at least the accuracy scores (Dice) of the AI-generated segmentation results over the ground truth.


--------

* **Training details and Computational Cost**: Mention the training specifics (hardware, learning rate, etc.) and also provide a quantitative comparison of the runtime of SACP versus standard CCCP.

- For the hyperparameter γ, it would be useful to know how varying γ would affect the coverage and other reported metrics.
----------

* **Compare to Other Uncertainty Methods**: Provide a direct comparison if possible. If not, please discuss how SACP's approach to uncertainty quantification would be advantageous over other methods (e.g., SACP by Liu et al. that has been cited as a reference in this paper; other ensemble methods).

---------

* **Clarify the baseline and variable results in Table 2** : For evaluating both the coverage and RWR values, please clarify on what were the baselines these were compared against (ex.: for coverage, the target seems to be 0.95).

In Table 2, it seems that the variable improvements for the different vessels (except CeTr) in all the other distance ranges (>2 mm) would eventually come down to the limited validation study done (n=30).

Building upon the first rebuttal point in this section, please provide more insights on the possible causes of this variability, if a larger validation size would change the results and if this may be due to the apparent lack of robustness of SACP.

-----------

* **Include references to papers** : Here are a few recent papers focusing on local spatial conformal prediction which seem to be relevant to the SACP method proposed in this paper. Please cite them wherever possible:

[1] Mao, H., Martin, R., & Reich, B. J. (2023). Valid Model-Free Spatial Prediction. Journal of the American Statistical Association, 119(546), 904–914. https://doi.org/10.1080/01621459.2022.2147531

[2] Leying Guan, Localized conformal prediction: a generalized inference framework for conformal prediction, Biometrika, Volume 110, Issue 1, March 2023, Pages 33–50, https://doi.org/10.1093/biomet/asac040

[3] Rina Foygel Barber, Emmanuel J. Candès, Aaditya Ramdas, Ryan J. Tibshirani "Conformal prediction beyond exchangeability," The Annals of Statistics, Ann. Statist. 51(2), 816-845, (April 2023)

-----------------

**Special Issue:**

No

---

> ### Author Response · Authors · 2025-03-08
> **Response to Reviewer Pciq (Part 1)**
>
> **C1:** Our validation spans two datasets: 60 scans total from the PANORAMA Challenge and MSK Medical Segmentation Decathlon. From PANORAMA's 2000+ scans, we filtered to 676 PDAC cases with 482 expert segmentations, then randomly selected 30 from ~150 eligible cases after excluding post treatment scans. While seemingly limited, each scan contains millions of voxels (512×512×200-400), providing robust statistical power across diverse clinical scenarios, patient populations, and tumor types.
>
> **C2:** We've incorporated SSCV analysis in Section 3.3 (Page 8) examining empirical coverage variation with prediction set cardinality.
>
> **C3:**  We have now included a comprehensive RWR analysis in Table 8 of Appendix M. The RWR values demonstrate that our spatially-adaptive approach maintains effective prediction set sizes while achieving improved coverage.
>
> **C4:**  We've included a comprehensive Dice coefficient analysis in Appendix H (Page 20) showing significantly reduced accuracy near vessel interfaces, with mean Dice scores decreasing by 12\% in PANORAMA and 38\% in MSK when near vessels in Tables 4 and 5, respectively. This performance gap underscores the importance of our spatially-adaptive uncertainty quantification approach.
>
> **C5:** Conformal prediction employs a calibration process rather than training. The computational cost difference between SACP and CCCP is negligible, with SACP adding only O(1) complexity operations. Implementation details and runtime comparisons are included in Appendix I (Page 21).
>
> **C6:**  Appendix J details our analysis of different $\gamma$ configurations. While all maintain strong coverage, they differ in spatial adaptation characteristics. We observed that $\gamma$ effectively controls the trade-off between coverage guarantees and spatial adaptivity. Clinical guidelines inform optimal values, with NCCN guidelines suggesting lower $\gamma$ for arterial vessels given their critical role in resectability assessment.
>
> **C7:**  SACP offers several advantages over existing uncertainty quantification methods. Unlike Liu et al.'s spatially-aware CP for hyperspectral imaging, our approach incorporates critical structure proximity and anatomical importance weights specific to medical contexts. Compared to ensemble methods (e.g., MC-Dropout, deep ensembles), SACP provides formal coverage guarantees without requiring multiple forward passes or model modifications. Ensemble methods cannot adapt their uncertainty estimates based on domain-specific anatomical relationships, whereas SACP explicitly incorporates this critical context. Traditional uncertainty methods also produce uniform confidence bounds across the entire image, while SACP generates targeted, conservative prediction sets specifically in regions that matter most for clinical decision-making. We've expanded our Related Work section to better position SACP in the landscape of uncertainty quantification methods while acknowledging that a direct empirical comparison remains an area for future work.
>
> **C8:** Our baseline is the standard 0.95 confidence level, with Table 2 comparing SACP against CCCP across vessel proximity zones.
>
> **C9:**  The variability in vessel-specific results primarily reflects natural anatomical complexity rather than statistical limitations. Different vessels have distinct spatial relationships with tumors—the celiac trunk maintains a more consistent superior-anterior relationship with the pancreas, while superior mesenteric vessels have complex circumferential relationships with tumors. These anatomical differences naturally produce variable prediction patterns across vessels. While we acknowledge our sample size limitations (30 cases per dataset), even with a larger dataset, we would expect to observe similar vessel-specific variations due to these inherent anatomical differences.
>
> **C10:**  We've integrated the suggested references on spatial conformal prediction in Page 3 (Section 1) including Mao et al. (2023) on model-free spatial prediction, Guan (2023) on localized conformal frameworks, and Barber et al. (2023) on conformal prediction beyond exchangeability.
>
> **C11:**  We've expanded the definition in Section 3.1 (Page 7) explaining that coverage rate's clinical relevance lies in its relationship to prediction conservativeness—higher coverage indicates where the model is more cautious, signaling to clinicians where additional diagnostic scrutiny may be warranted through further imaging or expert review.
>
> **C12:**  Section 3 **AI-Generated Segmentations** (Page 7) now clarifies that our PDAC model uses a tripartite architecture (teacher-professor-student) with 3D UNet cascades. The model was trained on 517 CT scans from clinical trials for the teacher model, expanding to 1085 CTs from 903 patients for the final model. All data included expert radiologist annotations and automated structure segmentations.

---

> > ### Comment · Reviewer_Pciq · 2025-03-08
> > **[To the PC, AC and other reviewers] regd supplementary material**
> >
> > Hello,
> >
> > I have a question regarding the supplementary material submitted which is close to 14 pages.
> >
> > 1. I have gone through the rebuttal by the authors and also reviewed the modifications they have made to this paper. While it has definitely improved - especially the related work and other clarifications regarding the methodology, the experiments still remain very limited. Going strictly by the guidelines, the reviewers have to consider only the main paper. However, if you read the rebuttal, a significant number of changes have been reported in the supplementary section (Appendix G - M).
> >
> >   -  Do we need to consider the results reported in the supplementary section for an overall assessment of the paper ?
> >
> >   -  The authors mention that they have validated this method additionally on the MSK dataset. However, only the PANORAMA dataset results have been reported in the main paper. Moreover, the MSK Results reported in the supplementary section confirm that segmentation quality is important towards making this method work. Do the reviewers have to report this too or simply ignore it since it isn't in the main paper ?
> >
> > On one hand the paper seems highly relevant especially for clinical applications especially in surgery. On the other hand, there is so much important material in the supplementary material, I cannot help but think if this will be better off as an extended journal submission.
> >
> > Looking forward to knowing what you think of my comments.

---

> > > ### Comment · Reviewer_R3Em · 2025-03-09
> > >
> > > Hi,
> > >
> > > I definitely see your concerns regarding the appendix and I agree. According to the instructions we don't have to go into the appendix but of course additional results, proofs, figures etc. add additional value. The paper should fully work without the appendix though, otherwise the page limit doesn't make sense. Whether it fulfills this criteria is up to you :)

---

> > > > ### Comment · Reviewer_Pciq · 2025-03-11
> > > >
> > > > Thanks for your comment. I think I will iterate on the paper once more with the authors. I see a lot of redundant information in the revised manuscript -  this can be made tighter and more concise. Maybe add the table including the MSK results too.
> > > >
> > > > Overall, the method is definitely novel and has a lot of impact potential - this may attract the attention of folks from within and outside the MIDL community. So I am more inclined to have this as part of the conference.

---

> ### Author Response · Authors · 2025-03-08
> **Response to Reviewer Pciq (Part 2)**
>
> **C13:** We've added a concise definition (Section 2 **Conformal Prediction**, Page 3) explaining how conservativeness ensures prediction sets maintain valid coverage guarantees by being slightly over-cautious rather than under-cautious in their coverage bounds.
>
> **C14:**  We've expanded Equation 4's description in Section 2.2 (Page 5) to clarify its intuitive pattern: weight decreases (approaching 0.5) when voxels are closer to vessels or when the model has high tumor confidence, creating more conservative sets in critical regions. Appendix J demonstrates robust behavior across different clinical scenarios and guideline requirements.
>
> **C15:** We evaluated against CCCP using Least Ambiguous Classifiers (LAC). Table 1 shows SACP achieves improved coverage while maintaining efficient prediction sets compared to this validated benchmark.

---

> > ### Comment · Reviewer_Pciq · 2025-03-11
> >
> > Thank you for addressing my comments and for sharing the revised manuscript.
> >
> > I also want to take this opportunity to remind you of the reviewer guidelines - while the supplementary material does add value in terms of more details, reviewers are supposed to evaluate only the main paper for the final decision, i.e. the 10-page main paper should be sufficient. Having said that, here are some of my additional comments, and a list of suggestions (it is entirely upto the authors' discretion to implement these):
> >
> > - the current version of the manuscript is MUCH better and clarifies a lot of ambiguity that the other reviewers had brought up.
> >
> > - please go through the manuscript once again -  while the changes are appreciated, there is still a lot of repetition and redundancy throughout the paper. This can be made more concise.
> >
> > - regarding your comment "each scan contains millions of voxels (512×512×200-400), providing robust statistical power across diverse clinical scenarios, patient populations, and tumor types." : fair point, but for the particular use-case you have given in this paper (i.e. tumor segmentation), the number of relevant voxels (i.e. ones belonging to the tumor) and their proximity to blood vessels is relatively smaller than 200-400. So in that context, n=20 seems to be a small validation set. Having more validation data gives a better representation of the robustness of the proposed method across different cases involving variations in tumor sizes and proximity to critical structures.
> >
> > - correct the title to "SACP: Spatially-Adaptive Conformal Prediction for Uncertainty Quantification in Medical Image Segmentation"
> >
> > - In Section 2, sub-section 'Conformal Prediction', introduce the concept of 'conservativeness' from the appendix section, i.e., revise the part "For a given significance level \alpha, ...... drawn from the same distribution". (Integrate the information from the appendix, "This property, known as conservativeness, guarantees that the probability of the true label being included in the prediction set is at least 1 − α, often making CP slightly over-conservative due to the discrete nature of rank-based p-values in finite samples.").
> >
> > - Sub-section 2.2.: modify "crucial in tumor segmentation task" to "crucial in the tumor segmentation task"
> >
> > - If after the modifications, space permits please add a table comparing the CCCP to SACP values for the MSK dataset.
> >
> > - Above Section 3, experimental evaluations, move the part "Segmentation models for PDAC... uncertainty quantification" to this section (in AI-Generated Segmentations)
> >
> > - IF you include the msk results in the main paper, in Section 3, modify the information in the sub-sections to include the MSK data pre-processing steps (in addition to the PDAC tumor content already present there).
> >
> >  These are my final suggestions. I would consider increasing my rating to 'Weak Accept' after the authors' response.

---

> > > ### Author Response · Authors · 2025-03-12
> > >
> > > Dear Reviewer Pciq,
> > > Thank you for your thoughtful feedback on our manuscript. We're pleased to hear that our revised version represents a significant improvement.
> > > We note that the title "SACP: Spatially-Adaptive Conformal Prediction" is already used in our current submission as you suggested.
> > > We have reviewed your other suggestions carefully and will implement the following changes:
> > > 1. We will address the redundancy you identified throughout the paper to make the content more concise and focused.
> > > 2. Regarding dataset validation, we'll add a clearer discussion of the statistical considerations. While our analysis leverages millions of voxels, we acknowledge your valid point about the more limited number of tumor-vessel interface cases. We'll add this nuance to our limitations section. In our future work section, we will also address expanding validation across a larger and more diverse collection of tumor-vessel interface cases.
> > > 3. To enhance the theoretical foundation presented in the main paper, we will integrate the concept of "conservativeness" from Appendix D into Section 2 (Conformal Prediction).
> > > 4. To improve the paper's logical flow, we will relocate the segmentation model description to the Experimental Evaluations section under AI-Generated Segmentations.
> > > Regarding the MSK dataset comparison table, we will examine our options for including this data in the main text while working within the page limit constraints. We will determine the best approach for presenting this information through a compact table or by incorporating key comparative metrics in the discussion.
> > > Thank you again for your constructive feedback and commitment to improving our work.
> > > Sincerely,
> > > Authors

---

> > > > ### Comment · Reviewer_Pciq · 2025-03-13
> > > >
> > > > Thank you for your comments. I have revised my rating to '4: Weak Accept'. I have no further comments to add.

---

### Author Response · Authors · 2025-03-08
**Summary of Rebuttal**

We appreciate the reviewers' thorough and insightful feedback on our manuscript. We believe addressing and incorporating these comments significantly improves our manuscript. Below we address the key concerns:

**Validation & Datasets:** Our method was validated across two distinct datasets (PANORAMA and MSK), totaling 60 CT scans from multiple medical centers. While this may seem limited, our voxel-wise analysis operates on millions of data points, providing robust statistical power. The second dataset contains different tumor types (resectable PDAC, IPMN, and PNETs), demonstrating generalizability across diverse clinical scenarios.

**Specificity of Approach:** While our method was motivated by pancreatic cancer, the principles apply broadly to situations with: (1) spatially structured prediction tasks, (2) canonical objects, and (3) critical structures. This framework transfers readily to domains like autonomous driving (pedestrian safety), industrial quality control (critical component inspection), and natural disaster prediction (risk assessment near populated areas).

**Methodology \& Experiments:**
- We clarified our workflow, visualization process, and implementation details.
- We expanded Table 2 to include RWR metrics and added SSCV analysis.
- As requested, we conducted additional experiments comparing SACP with uniformly conservative CCCP, demonstrating SACP's advantage in maintaining efficient prediction sets while ensuring coverage.
- We expanded our vessel importance weight ($\gamma$) analysis, showing SACP's adaptability to different clinical priorities.
- We updated references to include recent work on spatial CP applications.

**Theoretical Guarantees:** We provided clearer explanation of the mathematical guarantees in the main paper, particularly regarding conservativeness and spatial adaptivity, while keeping detailed proofs in the appendix.

**Limitations:** We expanded the discussion of limitations, acknowledging dependency on vessel segmentation quality and potential challenges in clinical deployment. We noted that vessel segmentation benefits from strong contrast in CT imaging, making this task more reliable than tumor segmentation. We also outlined future directions for clinical integration.

**Technical Improvements:**
- Fixed terminology consistency ("non-conformity" vs. "nonconformity")
- Corrected author affiliations format
- Added missing citations for CCCP (Shi et al. 2013, Sadinle et al. 2019)
- Updated related work to include Fischer et al. 2024, etc.

**Higher Precision with Lower Uncertainty:** We clarified that our method does not improve segmentation performance per se, but rather provides spatially-adaptive uncertainty quantification that focuses conservativeness where clinically needed.

Our revised manuscript offers a novel approach to uncertainty quantification in medical image segmentation that provides anatomically-informed prediction sets with valid coverage guarantees, addressing a critical need in surgical planning and other spatially structured prediction tasks.

---

### Author Rebuttal · Authors · 2025-03-08

**Rebuttal:**

The revised manuscript for this stage has been uploaded. We have included the revised/modified parts highlighted in red.

In summary, we have addressed all key concerns in the revised submission by:
- Clarifying our validation's robustness across diverse datasets
- Demonstrating broader applicability beyond pancreatic cancer
- Providing additional experiments that reinforce our method's advantages
- Improving the manuscript's clarity, terminology, and references

**Supporting Material:**

/attachment/51089d4ed50ef4d86ad46cd6cf8c6683e02ff068.pdf

---

### Meta-Review · Area_Chair_xLA6 · 2025-03-21

**Recommendation:** Accept (Poster)
**Confidence:** 4

**Metareview:**

The paper proposes a novel method SACP that addresses the limitations of standard Conformal Prediction (CP) in scenarios where different spatial regions require distinct certainty measurements. This can have a strong clinical impact especially in surgical interventions where the operability of a tumor is determined by the certainty in which the tumor and surrounding structures are detected.
The methodology outlined in this paper is very rigorous and backed with sound theoretical proofs. The revised version submitted by the authors after the rebuttal is more organized and addresses a lot of questions raised by the reviewers. There is still some weaknesses (using a small subset of the large public dataset "PANORAMA" for validating the method; having lot of redundant information in the methodology section). However, the merits of this paper outweigh these weaknesses.